# Improving Neural Network Generalization via Promoting Within-Layer Diversity

## Abstract

Neural networks are composed of multiple layers arranged in a hierarchical structure jointly trained with a gradient-based optimization, where the errors are back-propagated from the last layer back to the first one. At each optimization step, neurons at a given layer receive feedback from neurons belonging to higher layers of the hierarchy. In this paper, we propose to complement this traditional 'between-layer' feedback with additional 'within-layer' feedback to encourage the diversity of the activations within the same layer. To this end, we measure the pairwise similarity between the outputs of the neurons and use it to model the layer's overall diversity. By penalizing similarities and promoting diversity, we encourage each unit within the layer to learn a distinctive representation and, thus, to enrich the data representation learned and to increase the total capacity of the model. We derive novel generalization bounds for neural networks depending on the within-layer activation diversity and prove that increasing the diversity of hidden activations reduces the estimation error. In addition to the theoretical guarantees, we present an extensive empirical study confirming that the proposed approach enhances the performance of state-of-the-art neural network models and decreases the generalization gap in multiple tasks.

## 1 Introduction

Neural networks are a powerful class of non-linear function approximators that have been successfully used to tackle a wide range of problems. They have enabled breakthroughs in many tasks, such as image classification (Krizhevsky et al., 2012), speech recognition (Hinton et al., 2012a), and anomaly detection (Golan & El-Yaniv, 2018). Formally, the output of a neural network consisting of P layers can be defined as follows:

$$f(\boldsymbol{x}; \mathbf{W}) = \phi^P(\boldsymbol{W}^P(\phi^{P-1}(\cdots \phi^2(\boldsymbol{W}^2 \phi^1(\boldsymbol{W}^1 \boldsymbol{x}))))), \tag{1}$$

where $\phi^i(.)$ is the element-wise activation function, e.g., *ReLU* or *Sigmoid*, of the $i^{th}$ layer and $\mathbf{W} = \{\boldsymbol{W}^1, \ldots, \boldsymbol{W}^P\}$ are the corresponding weights of the network. The parameters of $f(\boldsymbol{x}; \mathbf{W})$ are optimized by minimizing the empirical loss:

$$\hat{L}(f) = \frac{1}{N} \sum_{i=1}^{N} l\big(f(\boldsymbol{x}_i; \mathbf{W}), y_i\big), \tag{2}$$

where $l(\cdot)$ is the loss function, and $\{\boldsymbol{x}_i, y_i\}_{i=1}^{N}$ are the training samples and their associated ground-truth labels. The loss is minimized using the gradient decent-based optimization coupled with back-propagation.

However, neural networks are often over-parameterized, i.e., have more parameters than data. As a result, they tend to overfit to the training samples and not generalize well on unseen examples (Goodfellow et al., 2016). While research on double descent (Belkin et al., 2019; Advani et al., 2020; Nakkiran et al., 2020) shows that over-parameterization does not necessarily lead to overfitting, avoiding overfitting has been extensively studied (Neyshabur et al., 2018; Nagarajan & Kolter, 2019; Poggio et al., 2017; Dziugaite & Roy, 2017; Foret et al., 2020) and various approaches and strategies have been proposed, such as data augmentation (Goodfellow et al., 2016; Zhang et al., 2018), regularization (Kukačka et al., 2017; Bietti et al., 2019; Arora et al., 2019), and Dropout

(Hinton et al., 2012b; Wang et al., 2019; Lee et al., 2019; Li et al., 2016), to close the gap between the empirical loss and the expected loss.

Diversity of learners is widely known to be important in ensemble learning (Li et al., 2012; Yu et al., 2011) and, particularly in deep learning context, diversity of information extracted by the network neurons has been recognized as a viable way to improve generalization (Xie et al., 2017a; 2015b). In most cases, these efforts have focused on making the set of weights more diverse (Yang et al.; Malkin & Bilmes, 2009). However, diversity of the activations has not received much attention. Here, we argue that due to the presence of non-linear activations, diverse weights do not guarantee diverse feature representation. Thus, we propose applying the diversity on top of feature mapping instead of the weights.

To the best of our knowledge, only Cogswell et al. (2016) have considered diversity of the activations directly in the neural network context. They proposed an additional loss term using cross-covariance of hidden activations, which encourages the neurons to learn diverse or non-redundant representations. Their proposed approach, known as DeCov, was empirically proven to alleviate overfitting and to improve the generalization ability of neural network, yet a theoretical analysis to prove this has so far been lacking. Moreover, modeling diversity as the sum of the pairwise cross-covariance, it can capture only the pairwise diversity between components and is unable to capture the "higher-order diversity".

In this work, we start by theoretically showing that the within-layer activation diversity boosts the generalization ability of neural networks and reduces overfitting. Moreover, we propose a novel approach to encourage activation diversity within a layer. We propose complementing the 'between-layer' feedback with additional 'within-layer' feedback to penalize similarities between neurons on the same layer. Thus, we encourage each neuron to learn a distinctive representation and to enrich the data representation learned within each layer. We propose three variants for our approach that are based on different global diversity definitions.

Our contributions in this paper are as follows:

- Theoretically, we derive novel generalization bounds for neural networks depending on the within-layer activation diversity. As shown in Section 2, we express the upper-bound of the estimation error as a function of the diversity factor. Thus, we provide theoretical evidence that the within-layer activation diversity helps reduce the generalization error.
- Methodologically, we propose a new approach to encourage the 'diversification' of the layers' output feature maps in neural networks. The proposed approach has three variants. The main intuition is that by promoting the within-layer activation diversity, neurons within a layer learn distinct patterns and, thus, increase the overall capacity of the model.
- Empirically, we show that the proposed within-layer activation diversification boosts the performance of neural networks. Experimental results on several tasks show that the proposed approach outperforms competing methods.

## 2 GENERALIZATION ERROR ANALYSIS

In this section, we derive generalization bounds of neural networks depending on the within-layer activation diversity. Generalization theory (Zhang et al., 2017; Kawaguchi et al., 2017) focuses on the relation between the empirical loss defined in equation 2 and the expected risk, for any $f$ in the hypothesis class $\mathcal{F}$, defined as follows:

$$L(f) = \mathbb{E}_{(\boldsymbol{x},y)\sim\mathcal{Q}}[l(f(\boldsymbol{x}), y)], \tag{3}$$

where $\mathcal{Q}$ is the underlying distribution of the dataset. Let $f^* = \arg\min_{f\in\mathcal{F}} L(f)$ be the expected risk minimizer and $\hat{f} = \arg\min_{f\in\mathcal{F}} \hat{L}(f)$ be the empirical risk minimizer. We are interested in the estimation error, i.e., $L(f^*) - L(\hat{f})$, defined as the gap in the loss between both minimizers (Barron, 1994). The estimation error represents how well an algorithm can learn. It usually depends on the complexity of the hypothesis class and the number of training samples (Barron, 1993; Zhai & Wang, 2018).

In this work, we are interested in the effect of the within-layer activation diversity on the estimation error. In order to study this effect, given a layer with M units, we assume that with a high proba-

bility $\tau$, the average pairwise distance between the outputs of the units, $\frac{1}{2M(M-1)} \sum_{i \neq j}^{M} (\phi_n(\boldsymbol{x}) - \phi_m(\boldsymbol{x}))^2$, is lower bounded by $d_{min}^2$ for any input $\boldsymbol{x}$. Intuitively, if two neurons $n$ and $m$ have similar outputs for many samples, the corresponding distance $(\phi_n(\boldsymbol{x}) - \phi_m(\boldsymbol{x}))^2$ will be small. If the average distance is small, the lower bound $d_{min}$ is also small and the units within this layer are considered redundant and "not diverse". Otherwise, if the average distance between the different pairs is large, their corresponding $d_{min}$ is large and they are considered "diverse". By studying how the lower bound $d_{min}$ affects the generalization of the model, we can theoretically understand how diversity affects the performance of neural networks. To this end, we derive generalization bounds for neural networks using $d_{min}$.

Several techniques have been used to quantify the estimation error, such as PAC learning (Shalev-Shwartz & Ben-David, 2014; Valiant, 1984), VC dimension (Sontag, 1998), and the Rademacher complexity (Shalev-Shwartz & Ben-David, 2014). The Rademacher complexity has been widely used as it usually leads to a tighter generalization error bound (Sokolic et al., 2016; Neyshabur et al., 2018; Golowich et al., 2018). The formal definition of the empirical Rademacher complexity is given as follows:

**Definition 1.** *(Shalev-Shwartz & Ben-David, 2014; Bartlett & Mendelson, 2002) For a given dataset with N samples $\mathcal{D} = \{\boldsymbol{x}_i, y_i\}_{i=1}^{N}$ generated by a distribution $\mathcal{Q}$ and for a model space $\mathcal{F} : \mathcal{X} \to \mathbb{R}$ with a single dimensional output, the empirical Rademacher complexity $\mathcal{R}_N(\mathcal{F})$ of the set $\mathcal{F}$ is defined as follows:*

$$\mathcal{R}_N(\mathcal{F}) = \mathbb{E}_\sigma \left[ \sup_{f \in \mathcal{F}} \frac{1}{N} \sum_{i=1}^{N} \sigma_i f(\boldsymbol{x}_i) \right], \tag{4}$$

*where the Rademacher variables $\sigma = \{\sigma_1, \cdots, \sigma_N\}$ are independent uniform random variables in $\{-1, 1\}$.*

In this work, we rely on the Rademacher complexity to study diversity. Our analysis starts with the following lemma:

**Lemma 1.** *(Bartlett & Mendelson, 2002) With a probability of at least $1 - \delta$*

$$L(\hat{f}) - L(f^*) \leq 4\mathcal{R}_N(\mathcal{A}) + B\sqrt{\frac{2\log(2/\delta)}{N}} \tag{5}$$

*where $B \geq \sup_{\boldsymbol{x},y,f} |l(f(\boldsymbol{x}), y)|$ and $\mathcal{R}_N(\mathcal{A})$ is the Rademacher complexity of the loss set $\mathcal{A}$.*

It upper-bounds the estimation error using the Rademacher complexity defined over the loss set and $\sup_{x,y,f} |l(f(x), y)|$. Our analysis continues by seeking a tighter upper bound of this error and showing how the within-layer diversity, expressed with $d_{min}$, affects the bound.

In this paper, we derive such an upper-bound for a simple network with one hidden layer trained for a regression task. We show how to extend it for classification, general multi-layer network and for different losses in the Appendix. The proofs are provided as supplementary material.

## 2.1 SINGLE HIDDEN-LAYER NETWORK

Here, we consider a simple neural network with one hidden-layer with $M$ neurons and one-dimensional output trained for a regression task. The full characterization of the setup can be summarized in the following assumptions:

**Assumptions 1.**

- *The input vector $\boldsymbol{x} \in \mathbb{R}^D$ satisfies $||\boldsymbol{x}||_2 \leq C_1$ and the output scalar $y \in \mathbb{R}$ satisfies $|y| \leq C_2$. The activation function of the hidden layer, $\phi(\cdot)$, is a positive $L_\phi$-Lipschitz continuous function.*

- *The weight matrix $\boldsymbol{W} = [\boldsymbol{w}_1, \boldsymbol{w}_2, \cdots, \boldsymbol{w}_M] \in \mathcal{R}^{D \times M}$ connecting the input to the hidden layer satisfies $||\boldsymbol{w}_m||_2 \leq C_3$. The weight vector $\boldsymbol{v} \in \mathbb{R}^M$ connecting the hidden-layer to the output neuron satisfies $||\boldsymbol{v}||_\infty \leq C_4$.*

- *The hypothesis class is $\mathcal{F} = \{f | f(\boldsymbol{x}) = \sum_{m=1}^{M} v_m \phi_m(\boldsymbol{x}) = \sum_{m=1}^{M} v_m \phi(\boldsymbol{w}_m^T \boldsymbol{x})\}$.*

- *Loss function set is $\mathcal{A} = \{l | l(f(\boldsymbol{x}), y) = \frac{1}{2}|f(\boldsymbol{x}) - y|^2\}$.*

- *With a probability $\tau$, $\frac{1}{2M(M-1)} \sum_{n \neq m}^{M} (\phi_n(\boldsymbol{x}) - \phi_m(\boldsymbol{x}))^2 \geq d_{min}^2$.*

The main idea of our proof is to find a diversity-dependant bound for both terms in Lemma 1. To this end, we derive a novel bound for loss hypothesis class $\mathcal{A}$ depending on $d_{min}$ and use it for both terms. The loss depends on the hypothesis class $\mathcal{F}$. Thus, we start by deriving an upper bound for this class:

**Lemma 2.** *Under Assumptions 1, with a probability at least $\tau$, we have*

$$\sup_{\boldsymbol{x}, f \in \mathcal{F}} |f(\boldsymbol{x})| \leq \sqrt{\mathcal{J}}, \tag{6}$$

*where $\mathcal{J} = C_4^2 \big( MC_5^2 + M(M-1)(C_5^2 - d_{min}^2) \big)$ and $C_5 = L_\phi C_1 C_3 + \phi(0)$,*

Note that in Lemma 2, we have expressed the upper-bound of $\sup_{\boldsymbol{x},f} |f(\boldsymbol{x})|$ in terms of $d_{min}$. Using this bound, we can now find an upper-bound for $\sup_{\boldsymbol{x},f,y} |l(f(\boldsymbol{x}), y)|$ in the following lemma:

**Lemma 3.** *Under Assumptions 1, with a probability at least $\tau$, we have*

$$\sup_{\boldsymbol{x},y,f} |l(f(\boldsymbol{x}), y)| \leq \frac{1}{2}(\sqrt{\mathcal{J}} + C_2)^2. \tag{7}$$

Our main goal is to analyze the estimation error bound of the neural network and to see how its upper-bound is linked to the diversity, expressed by $d_{min}$, of the different neurons. The main result of the paper is presented in Theorem 1.

**Theorem 1.** *Under Assumptions 1, there exist a constant A, such that with probability at least $\tau(1 - \delta)$, we have*

$$L(\hat{f}) - L(f^*) \leq \left(\sqrt{\mathcal{J}} + C_2\right) \frac{A}{\sqrt{N}} + \frac{1}{2}(\sqrt{\mathcal{J}} + C_2)^2 \sqrt{\frac{2\log(2/\delta)}{N}} \tag{8}$$

*where $\mathcal{J} = C_4^2 \big( MC_5^2 + M(M-1)(C_5^2 - d_{min}^2) \big)$, and $C_5 = L_\phi C_1 C_3 + \phi(0)$.*

Theorem 1 provides an upper-bound for the generalization gap. We note that it is a decreasing function of $d_{min}$. Thus, this suggests that higher $d_{min}$, i.e., more diverse activations, yields a lower estimation error bound. In other words, by promoting the within-layer diversity, we can reduce the generalization error of neural networks. It should be noted that in the last item of our assumptions, we considered a relaxed variant of the following assumption 'H*: There exists a lower bound to the distance, valid for any input **x**'. H* is impractical especially if the intermediate layer has ReLu activations. Therefore, we considered a relaxed variant of this assumption by introducing the probability $\tau$. This makes the theoretical findings useful in practice.

## 3 WITHIN-LAYER ACTIVATION DIVERSITY

As shown in the previous section, promoting diversity of activations within a layer can lead to tighter generalization bound and can theoretically decrease the gap between the empirical and the true risks. In this section, we propose a novel diversification strategy, where we encourage neurons within a layer to activate in a mutually different manner, i.e., to capture different patterns. To this end, we propose an additional within-layer loss which penalizes the neurons that activate similarly. The loss function $\hat{L}(f)$ defined in equation 2 is augmented as follows:

$$\hat{L}_{aug}(f) = \hat{L}(f) + \lambda J^i, \tag{9}$$

where $J^i$ expresses the overall similarity of the neurons within the $i^{th}$ layer and $\lambda$ is the penalty coefficient for the diversity loss. As in (Cogswell et al., 2016), our proposed diversity loss can be applied to a single layer or multiple layers in a network. For simplicity, let us focus on a single layer.

Let $\phi_n^i(\boldsymbol{x}_j)$ and $\phi_m^i(\boldsymbol{x}_j)$ be the outputs of the $n^{th}$ and $m^{th}$ neurons in the $i^{th}$ layer for the same input sample $\boldsymbol{x}_j$. The similarity $s_{nm}$ between the the $n^{th}$ and $m^{th}$ neurons can be obtained as the

average similarity measure of their outputs for $N$ input samples. We use the radial basis function to express the similarity:

$$s_{nm} = \frac{1}{N} \sum_{j=1}^{N} \exp\big( - \gamma ||\phi_n^i(\boldsymbol{x}_j) - \phi_m^i(\boldsymbol{x}_j)||^2 \big), \qquad (10)$$

where $\gamma$ is a hyper-parameter. The similarity $s_{nm}$ can be computed over the whole dataset or batch-wise. Intuitively, if two neurons $n$ and $m$ have similar outputs for many samples, their corresponding similarity $s_{nm}$ will be high. Otherwise, their similarity $s_{mn}$ is small and they are considered "diverse". Based on these pairwise similarities, we propose three variants for the overall similarity $J^i$ in equation 9:

- **Direct:** $J^i = \sum_{n \neq m} s_{nm}$. In this variant, we model the global layer similarity directly as the sum of the pairwise similarities between the neurons. By minimizing their sum, we encourage the neurons to learn different representations.

- **Det:** $J^i = -\det(\mathbf{S})$, where $\boldsymbol{S}$ is a similarity matrix defined as $\boldsymbol{S}_{nm} = s_{nm}$. This variant is inspired by the Determinantal Point Process (DPP) (Kulesza & Taskar, 2010; 2012), as the determinant of $\boldsymbol{S}$ measures the global diversity of the set. Geometrically, $\det(\boldsymbol{S})$ is the volume of the parallelepiped formed by vectors in the feature space associated with $s$. Vectors that result in a larger volume are considered to be more "diverse". Thus, maximizing $\det(\cdot)$ (minimizing $-\det(\cdot)$) encourages the diversity of the learned features.

- **Logdet:** $J^i = -\text{logdet}(\mathbf{S})^1$. This variant has the same motivation as the second one. We use logdet instead of det as logdet is a convex function over the positive definite matrix space.

It should be noted here that the first proposed variant, i.e., direct, similar to Decov (Cogswell et al., 2016), captures only the pairwise diversity between components and is unable to capture the higher-order "diversity", whereas the other two variants consider the global similarity and are able to measure diversity in a more global manner.

Our newly proposed loss function defined in equation 9 has two terms. The first term is the classic loss function. It computes the loss with respect to the ground-truth. In the back-propagation, this feedback is back-propagated from the last layer to the first layer of the network. Thus, it can be considered as a between-layer feedback, whereas the second term is computed within a layer. From equation 9, we can see that our proposed approach can be interpreted as a regularization scheme. However, regularization in deep learning is usually applied directly on the parameters, i.e., weights (Goodfellow et al., 2016; Kukačka et al., 2017), while in our approach, similar to (Cogswell et al., 2016), an additional term is defined over the output maps of the layers. For a layer with $C$ neurons and a batch size of $N$, the additional computational cost is $O(C^2(N + 1))$ for direct variant and $O(C^3 + C^2 N))$ for both the determinant and log of the determinant variants.

In connection to the bounds derived in Section 2 and to how diversity is theoretically defined in the last item in the Assumptions 1, we note that our regularizer relies on the pairwise RBF distance instead of the standard $L_2$ distance. We note that minimizing the average RBF distance, i.e., our direct variant, is equivalent to maximizing the average $L_2$ distance (exp(-x) is a decreasing function). Moreover, we note that empirically the RBF-based distance is less sensitive to noise (Savas & Dovis, 2019; Haykin, 2010).

## 4 RELATED WORK

**Diversity promoting strategies** have been widely used in ensemble learning (Li et al., 2012; Yu et al., 2011), sampling (Derezinski et al., 2019; Bıyık et al., 2019; Gartrell et al., 2019), energy-based models (Zhao et al., 2017; Laakom et al., 2021), ranking (Yang et al.; Gan et al., 2020), pruning by reducing redundancy (Kondo & Yamauchi, 2014; He et al., 2019; Singh et al., 2020; Lee et al., 2020), and semi-supervised learning (Zbontar et al., 2021). In the deep learning context, various approaches have used diversity as a direct regularizer on top of the weight parameters. Here,

---

[1]This is defined only if $\boldsymbol{S}$ is positive definite. It can be shown that in our case $\boldsymbol{S}$ is positive semi-definite. Thus, in practice we use a regularized version $(\boldsymbol{S} + \epsilon \boldsymbol{I})$ to ensure the positive definiteness.

we present a brief overview of these regularizers. Based on the way diversity is defined, we can group these approaches into two categories. The first group considers the regularizers that are based on the pairwise dissimilarity of components, i.e., the overall set of weights are diverse if every pair of weights are dissimilar. Given the weight vectors $\{w_m\}_{m=1}^M$, Yu et al. (2011) define the regularizer as $\sum_{mn}(1 - \theta_{mn})$, where $\theta_{mn}$ represents the cosine similarity between $w_m$ and $w_n$. Bao et al. (2013) proposed an incoherence score defined as $-\log\left(\frac{1}{M(M-1)}\sum_{mn}\beta|\theta_{mn}|^{\frac{1}{\beta}}\right)$, where $\beta$ is a positive hyperparameter. Xie et al. (2015a; 2016) used $\text{mean}(\theta_{mn}) - \text{var}(\theta_{mn})$ to regularize Boltzmann machines. They theoretically analyzed its effect on the generalization error bounds in (Xie et al., 2015b) and extended it to kernel space in (Xie et al., 2017a). The second group of regularizers considers a more globalist view of diversity. For example, in (Malkin & Bilmes, 2009; 2008; Xie et al., 2017b), a weight regularization based on the determinant of the weights' covariance is proposed based on determinantal point process in (Kulesza & Taskar, 2012; Kwok & Adams, 2012).

Unlike the aforementioned methods which promote diversity on the weight level and similar to our method, (Cogswell et al., 2016) proposed to enforce dissimilarity on the feature map outputs, i.e., on the activations. To this end, they proposed an additional loss based on the pairwise covariance of the activation outputs. Their additional loss, $L_{Decov}$ is defined as the squared sum of the non-diagonal elements of the global covariance matrix $C$ of the activations: $L_{Decov} = \frac{1}{2}(||C||_F^2 - ||\text{diag}(C)||_2^2)$, where $||.||_F$ is the Frobenius norm. Their approach, Decov, yielded superior empirical performance; however, it lacks theoretical proof. In this paper, we closed this gap and we showed theoretically how employing a diversity strategy on the network activations can indeed decrease the estimation error bound and improve the generalization of the model. Besides, we proposed variants of our approach which consider a global view of diversity.

## 5 EXPERIMENTAL RESULTS

### 5.1 IMAGE CLASSIFICATION

We start by evaluating our proposed diversity approach on two image datasets: CIFAR10 and CIFAR100 (Krizhevsky et al., 2009). We use our approach on three state-of-the-art CNNs: **ResNext-29-08-16**: we consider the standard ResNext Model (Xie et al., 2017c) with a 29-layer architecture, a cardinality of 8, and a width of 16. **DenseNet-12**: we use DenseNet (Huang et al., 2017) with the 40-layer architecture and a growth rate of 12. **ResNet50**: we consider the standard ResNet model (He et al., 2016) with 50 layers. We compare against the standard networks[2] as well as networks trained with DeCov diversity strategy (Cogswell et al., 2016). For each approach the hyperparameters are selected based on the validation set. The full experimental setup is presented in the Appendix. We report the average performance over three random seeds.

Table 1: Classification errors of the different approaches on CIFAR10 and CIFAR100 with three different models. Results are averaged over three random seeds.

| method | DenseNet-12 | | ResNext-29-08-16 | | ResNet50 | |
|---|---|---|---|---|---|---|
| | CIFAR10 | CIFAR100 | CIFAR10 | CIFAR100 | CIFAR10 | CIFAR100 |
| Standard | 7.07 | 29.25 | 6.93 | 26.73 | 8.27 | 34.06 |
| DeCov | 7.18 | 29.17 | 6.84 | 26.70 | 8.03 | 32.26 |
| Ours(direct) | **6.95** | 29.16 | 6.74 | **26.54** | 7.86 | 32.15 |
| Ours(det) | 7.04 | **28.78** | **6.67** | 26.67 | **7.73** | **32.12** |
| Ours(logdet) | 6.96 | 29.15 | 6.70 | 26.67 | 7.91 | 32.20 |

Table 1 reports the average top-1 errors of the different approaches with the three basis networks. We note that, compared to the standard approach, employing a diversity strategy generally boosts the results for all the three models and that our approach consistency outperforms both competing

---

[2]For the standard approach, the only difference is not using an additional diversity loss. The remaining regularizers, data augmentation, weight decay etc, are all applied as specified per-experiment.

Table 2: Performance of different models with different diversity strategies on ImageNet dataset

| Method | ResNet50 | | Wide-ResNet50 | |
| --- | --- | --- | --- | --- |
| | Top-1 Errors | Gap | Top-1 Errors | Gap |
| Standard | 23.84 | 2.87 | 22.42 | 6.33 |
| DeCov | **23.62** | 2.70 | 22.68 | 6.34 |
| Ours(direct) | 23.75 | 2.73 | 22.39 | 6.22 |
| Ours(det) | **23.62** | 2.77 | 22.33 | 6.13 |
| Ours(logdet) | 23.64 | **1.07** | **22.27** | **6.03** |

methods (standard and DeCov) in all the experiments. For example with ResNet50, the three variants of our proposed approach significantly reduce the test errors over both datasets: $0.36\% - 0.54\%$ improvement on CIFAR10 and $1.86\% - 1.94\%$ on CIFAR100.

**ImageNet:** To further demonstrate the effectiveness of our approach and its ability to reduce the generalization gap in neural networks, we conduct additional image classification experiments on the ImageNet-2012 classification dataset (Russakovsky et al., 2015) using the ResNet50 (He et al., 2016) and Wide-ResNet50 (Zagoruyko & Komodakis, 2016) models. The diversity term is applied on the last intermediate layer, i.e., the global average pooling layer for both DeCov and our method. For the hyperparameters, we use $\lambda = 0.001$ and $\gamma = 10$. The full experimental setup is presented in the Appendix. Table 2 reports the test errors of the different diversity strategies. To study the effect of diversity on the generalization gap, we also report the final training errors and the generalization gap, i.e., training accuracy - test accuracy. As it can be seen, diversity (our approach and DeCov) reduces the test error of the model and yields a better performance. We note that, in accordance with our theoretical findings in Section 2, using diversity indeed reduces overfitting and decreases the empirical generalisation gap of neural networks. In fact, our logdet variant reduces the empirical generalization gap of the model by $1.8\%$ compared to the standard approach. We note that our approach has a small additional time cost. For example for ResNet50, our direct, det and logdet variants take only $0.29\%$, $0.39\%$, and $0.49\%$ extra training time, respectively.

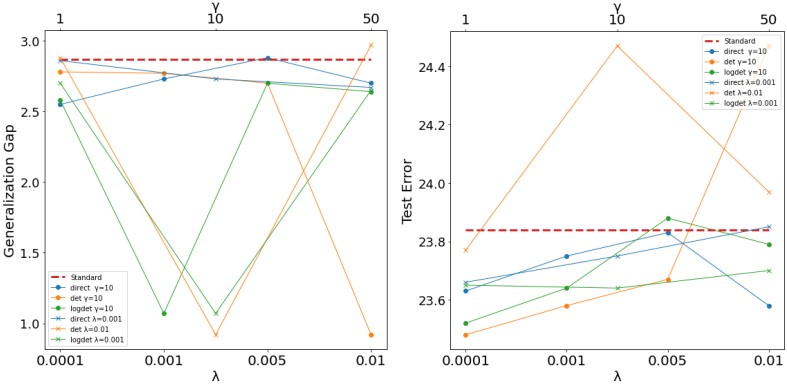

Figure 1: Sensitivity analysis of $\lambda$ and $\gamma$ on both the model accuracy and its generalization ability using ResNet50 trained on ImageNet.

**Sensitivity analysis:** To further investigate the effect of the proposed diversity strategy, we conduct a sensitivity analysis using ImageNet on the hyperparameters of our methods: $\gamma$, which is the RBF parameter used to measure the pairwise similarity between two units, and $\lambda$, which controls the contribution of the global diversity term to the global loss. We analyse the effect of the two parameters on both the final performance of the models and its generalization ability. The analysis is presented in Figure 1 and in supplementary material. As it can be seen, promoting the within-layer diversity consistently reduces overfitting and decreases the generalization gap for most of the hyperparameters values. Moreover, we note that global modeling of diversity, i.e., det and logdet variants, yield tighter generalization gaps compared to the non-global direct approach. It is worth noting that Figure 1 shows that there is a trade-off between the generalization gap and the final error. Emphasizing

Table 3: The compatibility of the proposed approach with Dropout. Test errors and generalization gap of different combinations on ImageNet dataset.

| Method | ResNet50 Top-1 Errors | Gap | Wide-ResNet50 Top-1 Errors | Gap |
|---|---|---|---|---|
| Dropout | 23.73 | 0.14 | 22.14 | 3.68 |
| Dropout + direct | 23.72 | **0.11** | 22.00 | **1.46** |
| Dropout + det | **23.65** | 0.26 | 22.16 | 3.80 |
| Dropout + logdet | **23.65** | 0.30 | **21.89** | 3.41 |

diversity and using a high weight for the diversity term significantly decreases the generalization gap, but this damages the performance of the model compared to the standard approach. For lower values of $\lambda$, the model is able to significantly outperform the standard approach on both the test error and the generalization gap.

**Connection to Theory**: In Section 2, we provided theoretical bounds for the generalization errors of neural networks, which are inversely proportional to the diversity term, $d_{min}$, the lower bound of $\frac{1}{2M(M-1)} \sum_{i \neq j}^{M} (\phi_n(\boldsymbol{x}) - \phi_m(\boldsymbol{x}))^2$. To show that the proposed regularizer indeed improves diversity, we track the empirical average of the aforementioned variable, i.e., $\frac{1}{2M(M-1)} \sum_{n \neq m}^{M} \sum_i (\phi_n(\boldsymbol{x}_i) - \phi_m(\boldsymbol{x}_i))^2$ during the training. The results for both the standard approach and the logdet variant of our approach are reported in Figure 2. As it can be seen, our regularizer yields in higher diversity which reduces overfitting and leads to better generalization.

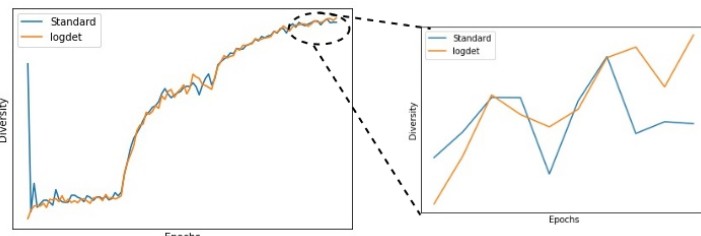

Figure 2: Tracking diversity in the training phase using ResNet50 trained on ImageNet.

**Compatibility with Dropout**: Dropout (Srivastava et al., 2014) is a popular regularization technique that, similarly to our approach, is applied on top of the layer output and has been known to improve generalization and reduce overfitting. Here, we evaluate the compatibility of our approach with Dropout. We add a Dropout regularizer with $20\%$ rate on top the last intermediate representation layer of both ResNet50 and Wide-ResNet50. The results are reported in Table 3. We note that adding a diversity regularizer alongside the Dropout consistently yields lower error rates compared to only Dropout except for the det variant on Wide-ResNet50. For Wide-ResNet50, the combination of our logdet variant and Dropout leads to $0.53\%$, $0.25\%$, and $0.38\%$ improvement compared to standard approach, Dropout only, and our logdet variant only, respectively.

**MLP-based models:** Beyond CNN models, we also evaluate the performance of our diversity strategy on modern attention-free, multi-layer perceptron (MLP) based models for image classification (Tolstikhin et al., 2021; Liu et al., 2021; Lee-Thorp et al., 2021). Such models are known to exhibit high overfitting and require regularization. We evaluate how diversity affects the accuracy of such models on CIFAR10. In particular, we conduct a simple experiment using two models: MLP-Mixer (Tolstikhin et al., 2021), gMLP (Liu et al., 2021) with four blocks each. The full description of the experimental setup is presented in the Appendix. The results in Table 4 show that employing a diversity strategy can indeed improve the performance of these models thanks to its ability to learn rich and robust representation of the input.

**Transfer learning:** Beyond standard classification, the proposed approach can be useful, e.g., in transfer learning, where the main goal is to 'transfer' previously learned representation to solve new

tasks. Thus, learning a rich and diverse representation is beneficial and can lead to better transferability. To demonstrate this, we conduct another experiment, where we use ImageNet-pretrained ResNet50 models with the different diversity approaches and we finetune them to CIFAR10 and CIFAR100. The results are reported in Table 5. As it can be seen, employing a diversity strategy helps in the transfer learning context and leads consistently to lower error rates. For example, the log variant of our approach leads to $0.94\%$ and $1.27\%$ gains on CIFAR10 and CIFAR100, respectively.

Table 4: Classification errors of modern MLP-based approaches on CIFAR10. Results are averaged over ten random seeds.

| | MLP-Mixer | gMLP |
|---|---|---|
| Standard | 23.96 | 24.69 |
| DeCov | 23.68 | 24.17 |
| Ours(direct) | 23.84 | **23.82** |
| Ours(det) | **23.62** | 24.08 |
| Ours(logdet) | 23.85 | 24.15 |

Table 5: Transfer learning performance on CIFAR10 and CIFAR100 of ResNet50 models pre-trained on ImageNet.

| | $\hookrightarrow$ CIFAR10 | $\hookrightarrow$ CIFAR100 |
|---|---|---|
| Standard | 6.14 | 22.99 |
| DeCov | 5.92 | 21.91 |
| Ours(direct) | 5.89 | **21.48** |
| Ours(det) | 5.51 | 22.01 |
| Ours(logdet) | **5.20** | 21.72 |

## 5.2 LEARNING IN THE PRESENCE OF LABEL NOISE

To further demonstrate the usefulness of promoting diversity, we test the robustness of our approach in the presence of label noise. In such situations, standard neural network tend to overfit to the noisy sample and not generalize well to the test set. Enforcing diversity can lead to better and richer representation attenuating the effect of noise. To show this, we performed additional experiments with label noise (20% and 40%) on CIFAR10 and CIFAR100 using ResNet50. The results are reported in Table 6. As it can be seen, in the presence of noise, the gap between the standard approach and diversity (Decov and ours) increases. For example, our logdet variant boosts the results by 1.71% and 3.59% on CIFAR10 and CIFAR100 with 40% noise, respectively.

Table 6: Classification errors of ResNet50 using different diversity strategies on CIFAR10 and CIFAR100 datasets with different label noise ratios. Results are averaged over three random seeds.

| | 20% label noise | | 40% label noise | |
|---|---|---|---|---|
| Method | CIFAR10 | CIFAR100 | CIFAR10 | CIFAR100 |
| Standard | 14.38 | 45.11 | 19.40 | 51.88 |
| DeCov | 13.64 | 43.02 | 18.11 | 50.66 |
| Ours(direct) | **13.48** | 41.78 | **17.57** | 48.91 |
| Ours(det) | 13.77 | 41.49 | 17.87 | 48.50 |
| Ours(logdet) | 13.59 | **41.11** | 17.69 | **48.29** |

## 6 CONCLUSIONS

In this paper, we proposed a new approach to encourage 'diversification' of the layer-wise feature map outputs in neural networks. The main motivation is that by promoting within-layer activation diversity, units within the same layer learn to capture mutually distinct patterns. We proposed an additional loss term that can be added on top of any fully-connected layer. This term complements the traditional 'between-layer' feedback with an additional 'within-layer' feedback encouraging diversity of the activations. We theoretically proved that the proposed approach decreases the estimation error bound and, thus, improves the generalization ability of neural networks. This analysis was further supported by extensive experimental results showing that such a strategy can indeed improve the performance of different state-of-the-art networks across different datasets and different tasks, i.e., image classification, transfer learning, and label noise. We are confident that these results will spark further research in diversity-based approaches to improve the generalization ability of neural networks.

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

## A   APPENDIX

The full characterization of the setup can be summarized in the following assumptions:

**Assumptions 2.**

- *The activation function of the hidden layer, $\phi(.)$, is a positive $L_\phi$-Lipschitz continuous function.*

- *The input vector $\boldsymbol{x} \in \mathbb{R}^D$ satisfies $||\boldsymbol{x}||_2 \leq C_1$.*

- *The output scalar $y \in \mathbb{R}$ satisfies $|y| \leq C_2$.*

- *The weight matrix $\boldsymbol{W} = [\boldsymbol{w}_1, \boldsymbol{w}_2, \cdots, \boldsymbol{w}_M] \in \mathcal{R}^{D \times M}$ connecting the input to the hidden layer satisfies $||\boldsymbol{w}_m||_2 \leq C_3$.*

- *The weight vector $\boldsymbol{v} \in \mathbb{R}^M$ connecting the hidden-layer to the output neuron satisfies $||\boldsymbol{v}||_\infty \leq C_4$.*

- *The hypothesis class is $\mathcal{F} = \{f | f(\boldsymbol{x}) = \sum_{m=1}^{M} v_m \phi_m(\boldsymbol{x}) = \sum_{m=1}^{M} v_m \phi(\boldsymbol{w}_m^T \boldsymbol{x})\}$.*

- *Loss function set is $\mathcal{A} = \{l | l(f(\boldsymbol{x}), y) = \frac{1}{2}|f(\boldsymbol{x}) - y|^2\}$.*

- *With a probability $\tau$, $\frac{1}{2M(M-1)} \sum_{n \neq m}^{M} (\phi_n(\boldsymbol{x}) - \phi_m(\boldsymbol{x}))^2 \geq d_{min}^2$.*

We recall the following two lemmas related to the estimation error and the Rademacher complexity:

**Lemma 4.** *(Bartlett & Mendelson, 2002) For $\mathcal{F} \in \mathbb{R}^{\mathcal{X}}$, assume that $g : \mathbb{R} \to \mathbb{R}$ is a $L_g$-Lipschitz continuous function and $\mathcal{A} = \{g \circ f : f \in \mathcal{F}\}$. Then we have*

$$\mathcal{R}_N(\mathcal{A}) \leq L_g \mathcal{R}_N(\mathcal{F}). \tag{11}$$

**Lemma 5.** *(Xie et al., 2015b) Under Assumptions 1, the Rademacher complexity $\mathcal{R}_N(\mathcal{F})$ of the hypothesis class $\mathcal{F} = \{f | f(\boldsymbol{x}) = \sum_{m=1}^{M} v_m \phi_m(\boldsymbol{x}) = \sum_{m=1}^{M} v_m \phi(\boldsymbol{w}_m^T \boldsymbol{x})\}$ can be upper-bounded as follows:*

$$\mathcal{R}_N(\mathcal{F}) \leq \frac{2L_\phi C_{134} M}{\sqrt{N}} + \frac{C_4 |\phi(0)| M}{\sqrt{N}}, \tag{12}$$

*where $C_{134} = C_1 C_3 C_4$ and $\phi(0)$ is the output of the activation function at the origin.*

### A.1   PROOF OF LEMMA 2

**Lemma 2** *Under Assumptions 1, with a probability at least $\tau$, we have*

$$\sup_{\boldsymbol{x}, f} |f(\boldsymbol{x})| \leq \sqrt{\mathcal{J}}, \tag{13}$$

*where $\mathcal{J} = C_4^2 \big( M C_5^2 + M(M-1)(C_5^2 - d_{min}^2) \big)$ and $C_5 = L_\phi C_1 C_3 + \phi(0)$.*

*Proof.*

$$f^2(\boldsymbol{x}) = \left( \sum_{m=1}^{M} v_m \phi_m(\boldsymbol{x}) \right)^2 \leq \left( \sum_{m=1}^{M} ||\boldsymbol{v}||_\infty \phi_m(\boldsymbol{x}) \right)^2 = ||\boldsymbol{v}||_\infty^2 \left( \sum_{m=1}^{M} \phi_m(\boldsymbol{x}) \right)^2 \leq C_4^2 \left( \sum_{m=1}^{M} \phi_m(\boldsymbol{x}) \right)^2$$

$$= C_4^2 \left( \sum_{m,n} \phi_m(\boldsymbol{x}) \phi_n(\boldsymbol{x}) \right) = C_4^2 \left( \sum_m \phi_m(\boldsymbol{x})^2 + \sum_{m \neq n} \phi_n(\boldsymbol{x}) \phi_m(\boldsymbol{x}) \right). \tag{14}$$

We have $\sup_{w,\boldsymbol{x}} \phi_m(\boldsymbol{x}) \leq \sup(L_\phi |\boldsymbol{w}^T \boldsymbol{x}| + \phi(0))$ because $\phi$ is $L_\phi$-Lipschitz. Thus, $||\phi||_\infty \leq L_\phi C_1 C_3 + \phi(0) = C_5$. For the first term in equation 14, we have $\sum_m \phi_m(\boldsymbol{x})^2 < M(L_\phi C_1 C_3 + \phi(0))^2 = M C_5^2$. The second term, using the identity $\phi_m(\boldsymbol{x}) \phi_n(\boldsymbol{x}) = \frac{1}{2}\big(\phi_m(\boldsymbol{x})^2 + \phi_n(\boldsymbol{x})^2 - (\phi_m(\boldsymbol{x}) - \phi_n(\boldsymbol{x}))^2\big)$, can be rewritten as

$$\sum_{m \neq n} \phi_m(\boldsymbol{x}) \phi_n(\boldsymbol{x}) = \frac{1}{2} \left( \sum_{m \neq n} \phi_m(\boldsymbol{x})^2 + \phi_n(\boldsymbol{x})^2 - \Big(\phi_m(\boldsymbol{x}) - \phi_n(\boldsymbol{x})\Big)^2 \right). \tag{15}$$

In addition, we have with a probability $\tau$, $\frac{1}{2}\sum_{m\neq n}(\phi_m(\boldsymbol{x}) - \phi_n(\boldsymbol{x}))^2 \geq M(M-1)d_{min}^2$ for $m \neq n$. Thus, we have with a probability at least $\tau$:

$$\sum_{m\neq n}\phi_m(\boldsymbol{x})\phi_n(\boldsymbol{x}) \leq \frac{1}{2}\sum_{m\neq n}(2C_5^2) - M(M-1)d_{min}^2 = M(M-1)(C_5^2 - d_{min}^2). \tag{16}$$

By putting everything back to equation 14, we have with a probability $\tau$,

$$f^2(\boldsymbol{x}) \leq C_4^2\Big(MC_5^2 + M(M-1)(C_5^2 - d_{min}^2)\Big) = \mathcal{J}. \tag{17}$$

Thus, with a probability $\tau$,

$$\sup_{\boldsymbol{x},f}|f(\boldsymbol{x})| \leq \sqrt{\sup_{\boldsymbol{x},f}f(\boldsymbol{x})^2} \leq \sqrt{\mathcal{J}}. \tag{18}$$

$\square$

## A.2 PROOF OF LEMMA 3

**Lemma 3** *Under Assumptions 1, with a probability at least $\tau$, we have*

$$\sup_{\boldsymbol{x},y,f}|l(f(\boldsymbol{x}),y)| \leq \frac{1}{2}(\sqrt{\mathcal{J}} + C_2)^2 \tag{19}$$

*Proof.* We have $\sup_{\boldsymbol{x},y,f}|f(\boldsymbol{x}) - y| \leq \sup_{\boldsymbol{x},y,f}(|f(\boldsymbol{x})| + |y|) = \sqrt{\mathcal{J}} + C_2$. Thus $\sup_{x,y,f}|l(f(x),y)| \leq \frac{1}{2}(\sqrt{\mathcal{J}} + C_2)^2$. $\square$

## A.3 PROOF OF THEOREM 1

**Theorem 1** *Under Assumptions 1, there exist a constant A, such that with probability at least $\tau(1-\delta)$, we have*

$$L(\hat{f}) - L(f^*) \leq \Big(\sqrt{\mathcal{J}} + C_2\Big)\frac{A}{\sqrt{N}} + \frac{1}{2}(\sqrt{\mathcal{J}} + C_2)^2\sqrt{\frac{2\log(2/\delta)}{N}} \tag{20}$$

*where $\mathcal{J} = C_4^2\big(MC_5^2 + M(M-1)(C_5^2 - d_{min}^2)\big)$, and $C_5 = L_\phi C_1 C_3 + \phi(0)$.*

*Proof.* Given that $l(\cdot)$ is $K$-Lipschitz with a constant $K = sup_{\boldsymbol{x},y,f}|f(\boldsymbol{x}) - y| \leq \sqrt{\mathcal{J}} + C_2$, and using Lemma 4, we can show that $\mathcal{R}_N(\mathcal{A}) \leq K\mathcal{R}_N(\mathcal{F}) \leq (\sqrt{\mathcal{J}} + C_2)\mathcal{R}_N(\mathcal{F})$. For $\mathcal{R}_N(\mathcal{F})$, we use the bound found in Lemma 5. Using Lemmas 1 and 3, we have

$$L(\hat{f}) - L(f^*) \leq 4\Big(\sqrt{\mathcal{J}} + C_2\Big)\Big(2L_\phi C_{134} + C_4|\phi(0)|\Big)\frac{M}{\sqrt{N}} + \frac{1}{2}(\sqrt{\mathcal{J}} + C_2)^2\sqrt{\frac{2\log(2/\delta)}{N}} \tag{21}$$

*where $C_{134} = C_1 C_3 C_4$, $\mathcal{J} = C_4^2\big(MC_5^2 + M(M-1)(C_5^2 - d_{min}^2)\big)$, and $C_5 = L_\phi C_1 C_3 + \phi(0)$. Thus, setting $A = 4\Big(2L_\phi C_{134} + C_4|\phi(0)|\Big)M$ completes the proof.* $\square$

## B BINARY CLASSIFICATION

We now extend our analysis of the effect of the within-layer diversity on the generalization error in the case of a binary classification task, i.e., $y \in \{-1, 1\}$. The extensions of Theorem 1 in the case of a hinge loss and a logistic loss are presented in Theorems 2 and 3, respectively.

**Theorem 2.** *Using the hinge loss, there exist a constant A, such that with probability at least $\tau(1 - \delta)$, we have*

$$L(\hat{f}) - L(f^*) \leq A/\sqrt{N} + (1 + \sqrt{\mathcal{J}})\sqrt{\frac{2\log(2/\delta)}{N}} \tag{22}$$

*where $\mathcal{J} = C_4^2(MC_5^2 + M(M-1)(C_5^2 - d_{min}^2))$ and $C_5 = L_\phi C_1 C_3 + \phi(0)$.*

**Theorem 3.** *Using the logistic loss $l(f(x), y) = \log(1 + e^{-yf(x)})$, there exist a constant A such that, with probability at least $\tau(1 - \delta)$, we have*

$$L(\hat{f}) - L(f^*) \leq \frac{A}{(1 + e^{\sqrt{-\mathcal{J}}})\sqrt{N}} + \log(1 + e^{\sqrt{\mathcal{J}}})\sqrt{\frac{2\log(2/\delta)}{N}} \quad (23)$$

*where $\mathcal{J} = C_4^2(MC_5^2 + M(M - 1)(C_5^2 - d_{min}^2))$ and $C_5 = L_\phi C_1 C_3 + \phi(0)$.*

As we can see, also for the binary classification task, the error bounds of the estimation error for the hinge and logistic losses are decreasing with respect to $d_{min}$. Thus, employing a diversity strategy can improve the generalization also for the binary classification task.

### B.1 Proofs of Theorems 2 and 3

Similar to the proofs of Lemmas 7 and 8 in Xie et al. (2015b), we can show the following two lemmas:

**Lemma 6.** *Using the hinge loss, we have with probability at least $\tau(1 - \delta)$*

$$L(\hat{f}) - L(f^*) \leq 4\left(2L_\phi C_{134} + C_4|\phi(0)|\right)\frac{M}{\sqrt{N}} + (1 + \sqrt{\mathcal{J}})\sqrt{\frac{2\log(2/\delta)}{N}} \quad (24)$$

*where $C_{134} = C_1 C_3 C_4$, $\mathcal{J} = C_4^2(MC_5^2 + M(M - 1)(C_5^2 - d_{min}^2))$, and $C_5 = L_\phi C_1 C_3 + \phi(0)$.*

**Lemma 7.** *Using the logistic loss $l(f(x), y) = \log(1 + e^{-yf(x)})$, we have with probability at least $\tau(1 - \delta)$*

$$L(\hat{f}) - L(f^*) \leq \frac{4}{1 + e^{\sqrt{-\mathcal{J}}}}\left(2L_\phi C_{134} + C_4|\phi(0)|\right)\frac{M}{\sqrt{N}} + \log(1 + e^{\sqrt{\mathcal{J}}})\sqrt{\frac{2\log(2/\delta)}{N}} \quad (25)$$

*where $C_{134} = C_1 C_3 C_4$, $\mathcal{J} = C_4^2(MC_5^2 + M(M - 1)(C_5^2 - d_{min}^2))$, and $C_5 = L_\phi C_1 C_3 + \phi(0)$.*

Taking $A = 4\left(2L_\phi C_{134} + C_4|\phi(0)|\right)M$ in Lemma 6 and Lemma 7 completes the proofs.

## C Multi-layer networks

Here, we extend our result for networks with P $(> 1)$ hidden layers. We assume that the pairwise distances between the activations within layer $p$ are lower-bounded by $d_{min}^{(p)}$ with a probability $\tau^{(p)}$. In this case, the hypothesis class can be defined recursively. In addition, we replace the fourth assumption in Assumptions 1 with: $||W^{(p)}||_\infty \leq C_3^{(p)}$ for every $W^{(p)}$, i.e., the weight matrix of the $p$-th layer. In this case, the main theorem is extended as follows:

**Theorem 4.** *There exist a constant A such that, with probability of at least $\prod_{p=0}^{P-1}(\tau^{(p)})(1 - \delta)$, we have*

$$L(\hat{f}) - L(f^*) \leq (\sqrt{\mathcal{J}^P} + C_2)\frac{A}{\sqrt{N}} + \frac{1}{2}\left(\sqrt{\mathcal{J}^P} + C_2\right)^2\sqrt{\frac{2\log(2/\delta)}{N}} \quad (26)$$

*where $\mathcal{J}^P$ is defined recursively using the following identities: $\mathcal{J}^0 = C_3^0 C_1$ and $\mathcal{J}^{(p)} = M^{(p)}C^{p2}\left(M^{p2}(L_\phi \mathcal{J}^{p-1} + \phi(0))^2 - M(M - 1)d_{min}^{(p)}{}^2\right)$, for $p = 1, \ldots, P$.*

In Theorem 4, we see that $\mathcal{J}^P$ is decreasing with respect to $d_{min}^{(p)}$. Thus, we see that maximizing the within-layer diversity, we can reduce the estimation error of a multi-layer neural network.

### C.1 Proof of Theorem 4

**Theorem 4** There exist a constant A such that, with probability of at least $\prod_{p=0}^{P-1}(\tau^{(p)})(1 - \delta)$, we have

$$L(\hat{f}) - L(f^*) \leq (\sqrt{\mathcal{J}^P} + C_2)\frac{A}{\sqrt{N}} + \frac{1}{2}\left(\sqrt{\mathcal{J}^P} + C_2\right)^2\sqrt{\frac{2\log(2/\delta)}{N}}, \quad (27)$$

where $\mathcal{J}^P$ is defined recursively using the following identities: $\mathcal{J}^0 = C_3^0 C_1$ and $\mathcal{J}^{(p)} = M^{(p)} C^{p2} \big( M^{p2} (L_\phi \mathcal{J}^{p-1} + \phi(0))^2 - M(M-1) d_{min}^{(p)}{}^2) \big)$, for $p = 1, \ldots, P$.

*Proof.* Lemma 5 in Xie et al. (2015b) provides an upper-bound for the hypothesis class. We denote by $\boldsymbol{v}^{(p)}$ the outputs of the $p^{th}$ hidden layer before applying the activation function:

$$\boldsymbol{v}^0 = [\boldsymbol{w}_1^{0^T} \boldsymbol{x}, ...., \boldsymbol{w}_{M^0}^{0^T} \boldsymbol{x}] \tag{28}$$

$$\boldsymbol{v}^{(p)} = [\sum_{j=1}^{M^{p-1}} w_{j,1}^{(p)} \phi(\boldsymbol{v}_j^{p-1}), ...., \sum_{j=1}^{M^{p-1}} w_{j,M^{(p)}}^{(p)} \phi(v_j^{p-1})] \tag{29}$$

$$\boldsymbol{v}^{(p)} = [\boldsymbol{w}_1^{(p)^T} \boldsymbol{\phi}^{(p)}, ..., \boldsymbol{w}_{M^{(p)}}^{(p)^T} \boldsymbol{\phi}^{(p)}], \tag{30}$$

where $\boldsymbol{\phi}^{(p)} = [\phi(v_1^{p-1}), \cdots, \phi(v_{M^{p-1}}^{p-1})]$. We have

$$||\boldsymbol{v}^{(p)}||_2^2 = \sum_{m=1}^{M^{(p)}} (\boldsymbol{w}_m^{(p)^T} \boldsymbol{\phi}^{(p)})^2 \tag{31}$$

and $\boldsymbol{w}_m^{(p)^T} \boldsymbol{\phi}^{(p)} \le C_3^{(p)} \sum_n \phi_n^{(p)}$. Thus,

$$||\boldsymbol{v}^{(p)}||_2^2 \le \sum_{m=1}^{M^{(p)}} (C_3^{(p)} \sum_n \phi_n^{(p)})^2 = M^{(p)} C_3^{p2} (\sum_n \phi_n^{(p)})^2 = M^{(p)} C_3^{p2} \sum_{mn} \phi_m^{(p)} \phi_n^{(p)}. \tag{32}$$

We use the same decomposition trick of $\phi_m^{(p)} \phi_n^{(p)}$ as in the proof of Lemma 2. We need to bound $\sup_x \phi^{(p)}$:

$$\sup_x \phi^{(p)} < \sup(L_\phi |\boldsymbol{v}^{p-1}| + \phi(0)) < L_\phi ||\boldsymbol{v}^{p-1}||_2^2 + \phi(0). \tag{33}$$

Thus, we have

$$||\boldsymbol{v}^{(p)}||_2^2 \le M^{(p)} C^{p2} \big( M^2 (L_\phi ||\boldsymbol{v}^{p-1}||_2^2 + \phi(0))^2 - M(M-1) d_{min}^2) \big) = \mathcal{J}^P. \tag{34}$$

We found a recursive bound for $||\boldsymbol{v}^{(p)}||_2^2$, we note that for $p = 0$, we have $||\boldsymbol{v}^0||_2^2 \le ||W^0||_\infty C_1 \le C_3^0 C_1 = \mathcal{J}^0$. Thus,

$$\sup_{\boldsymbol{x}, f^P \in \mathcal{F}^P} |f(\boldsymbol{x})| = \sup_{\boldsymbol{x}, f^P \in \mathcal{F}^P} |\boldsymbol{v}^P| \le \sqrt{\mathcal{J}^P}. \tag{35}$$

By replacing the variables in Lemma 1, we have

$$L(\hat{f}) - L(f^*) \le 4(\sqrt{\mathcal{J}^P} + C_2) \left( \frac{(2L_\phi)^P C_1 C_3^0}{\sqrt{N}} \prod_{p=0}^{P-1} \sqrt{M^{(p)}} C_3^{(p)} + \frac{|\phi(0)|}{\sqrt{N}} \sum_{p=0}^{P-1} (2L_\phi)^{P-1-p} \prod_{j=p}^{P-1} \sqrt{M^j} C_3^j \right)$$

$$+ \frac{1}{2} \left( \sqrt{\mathcal{J}^P} + C_2 \right)^2 \sqrt{\frac{2 \log(2/\delta)}{N}}$$

Taking $A = 4 * ((2L_\phi)^P C_1 C_3^0 \prod_{p=0}^{P-1} \sqrt{M^{(p)}} C_3^{(p)} + |\phi(0)| \sum_{p=0}^{P-1} (2L_\phi)^{P-1-p} \prod_{j=p}^{P-1} \sqrt{M^j} C_3^j)$ completes the proof. □

# D MULTIPLE OUTPUTS

Finally, we consider the case of a neural network with a multi-dimensional output, i.e., $\boldsymbol{y} \in R^D$. In this case, we can extend Theorem 1 with the following two theorems:

**Theorem 5.** *For a multivariate regression trained with the squared error, there exist a constant A such that, with probability at least $\tau(1 - \delta)$, we have*

$$L(\hat{f}) - L(f^*) \le (\sqrt{\mathcal{J}} + C_2) \frac{A}{\sqrt{N}} + \frac{D}{2} (\sqrt{\mathcal{J}} + C_2)^2 \sqrt{\frac{2 \log(2/\delta)}{N}} \tag{36}$$

*where $\mathcal{J} = C_4^2 (M C_5^2 + M(M-1)(C_5^2 - d_{min}^2))$ and $C_5 = L_\phi C_1 C_3 + \phi(0)$.*

**Theorem 6.** *For a multi-class classification task using the cross-entropy loss, there exist a constant A such that, with probability at least $\tau(1 - \delta)$, we have*

$$L(\hat{f}) - L(f^*) \leq \frac{A}{(D - 1 + e^{-2\sqrt{\mathcal{J}}})\sqrt{N}} + \log\left(1 + (D-1)e^{2\sqrt{\mathcal{J}}}\right)\sqrt{\frac{2\log(2/\delta)}{N}} \quad (37)$$

*where $\mathcal{J} = C_4^2(MC_5^2 + M(M-1)(C_5^2 - d_{min}^2))$ and $C_5 = L_\phi C_1 C_3 + \phi(0)$.*

Theorems 5 and 6 extend our result for the multi-dimensional regression and classification tasks, respectively. Both bounds are inversely proportional to the diversity factor $d_{min}$. We note that for the classification task, the upper-bound is exponentially decreasing with respect to $d_{min}$. This shows that increasing diversity within the layer yields a tighter generalization gap and, thus, theoretically guarantees a stronger generalization performance.

### D.1 PROOF OF THEOREM 5

*Proof.* The squared loss $\frac{1}{2}||f(\boldsymbol{x}) - \boldsymbol{y}||_2^2$ can be decomposed into D terms $\frac{1}{2}(f(\boldsymbol{x})_k - y_k)^2$. Using Theorem 1, we can derive the bound for each term and thus, we have:

$$L(\hat{f}) - L(f^*) \leq 4D(\sqrt{\mathcal{J}} + C_2)\left(2L_\phi C_{134} + C_4|\phi(0)|\right)\frac{M}{\sqrt{N}} + \frac{D}{2}(\sqrt{\mathcal{J}} + C_2)^2\sqrt{\frac{2\log(2/\delta)}{N}}, \quad (38)$$

where $C_{134} = C_1 C_3 C_4$, $\mathcal{J} = C_4^2(MC_5^2 + M(M-1)(C_5^2 - d_{min}^2))$, and $C_5 = L_\phi C_1 C_3 + \phi(0)$. Taking $A = 4D\left(2L_\phi C_{134} + C_4|\phi(0)|\right)M$ completes the proof.

$\square$

### D.2 PROOF OF THEOREM 6

*Proof.* Using Lemma 9 in Xie et al. (2015b), we have $\sup_{f,\boldsymbol{x},\boldsymbol{y}} l = \log\left(1 + (D-1)e^{2\sqrt{\mathcal{J}}}\right)$ and $l$ is $\frac{D-1}{D-1+e^{-2\sqrt{\mathcal{J}}}}$-Lipschitz. Thus, using the decomposition property of the Rademacher complexity, we have

$$\mathcal{R}_n(\mathcal{A}) \leq \frac{4D(D-1)}{D - 1 + e^{-2\sqrt{\mathcal{J}}}}\left(2L_\phi C_{134} + C_4|\phi(0)|\right)\frac{M}{\sqrt{N}}. \quad (39)$$

Taking $A = 4D(D-1)\left(2L_\phi C_{134} + C_4|\phi(0)|\right)M$ completes the proof. $\square$

## E EXPERIMENTAL RESULTS

Here, we report the experimental setup of the different experiments done in the paper along with a more detailed analysis of the results.

### E.1 CIFAR10 & CIFAR100

We start by evaluating our proposed diversity approach on two image datasets: CIFAR10 and CI-FAR100 (Krizhevsky et al., 2009). They contain 60,000 (50,000 train/10,000 test) $32 \times 32$ images grouped into 10 and 100 distinct categories, respectively. We split the original training set (50,000) into two sets: we use the first 40,000 images as the main training set and the last 10,000 as a validation set for hyperparameters optimization. We use our approach on three state-of-the-art CNNs: **ResNext-29-08-16**: we consider the standard ResNext Model (Xie et al., 2017c) with a 29-layer architecture, a cardinality of 8, and a width of 16. **DenseNet-12**: we use DenseNet (Huang et al., 2017) with the 40-layer architecture and a growth rate of 12. **ResNet50**: we consider the standard ResNet model (He et al., 2016) with 50 layers. We compare against the standard networks as well networks trained with DeCov diversity strategy (Cogswell et al., 2016).

All the models are trained using stochastic gradient descent (SGD) with a momentum of 0.9, weight decay of 0.0001, and a batch size of 128 for 200 epochs. The initial learning rate is set to 0.1 and is then decreased by a factor of 5 after 60, 120, and 160 epochs, respectively. We also adopt a standard data augmentation scheme that is widely used for these two datasets

(He et al., 2016; Huang et al., 2017). For all models, the additional diversity term is applied on top the last intermediate layer. For the hyperparameters: The loss weight is chosen from $\{0.00001, 0.00005, 0.0001, 0.0005, 0.001, 0.005, 0.01\}$ for both our approach and Decov and $\gamma$ in the radial basis function is chosen from $\{0.01, 0.1.1, 10, 50, 100\}$. For each approach, the model with the best validation performance is used in the test phase. We report the average performance over three random seeds.

### E.2    IMAGENET AND SENSITIVITY ANALYSIS

To further demonstrate the effectiveness of our approach and its ability to reduce the generalization gap in neural networks, we conduct additional image classification experiments on the ImageNet-2012 classification dataset (Russakovsky et al., 2015) using the ResNet50 model (He et al., 2016). The diversity term is applied on the last intermediate layer, i.e., the global average pooling layer for both DeCov and our method. For the hyperparameters, we use $\gamma = 10$ and $\lambda = 0.0001$ for all the different approaches. We use the standard augmentation practice for this dataset as in (Zhang et al., 2018; Huang et al., 2017; Cogswell et al., 2016). All the models are trained with a batch size of 256 for 100 epoch using SGD with Nesterov Momentum of 0.9. The learning rate is initially set to 0.1 and decreases at epochs 30, 60, 90 by a factor of 10.

To further investigate the effect of the proposed diversity strategy, we conduct a sensitivity analysis using ImageNet on the hyperparameters of our methods: $\gamma$, which is the RBF parameter used to measure the pairwise similarity between two units, and $\lambda$, which controls the contribution of the global diversity term to the global loss. We analyse the effect of the two parameters on both the final performance of the models and its generalization ability. The analysis is presented in Figure 1.

In Table 7, we report the same results of the main figure of the paper in a tabular form along with extra results. As it can be seen, promoting the within-layer diversity consistently reduces overfitting and decreases the generalization gap for most of the hyperparameters values. Moreover, we note that global modeling of diversity, i.e., det and logdet variants, yield tighter generalization gaps between the train and test errors compared to the non-global direct approach. In fact, while direct variant decreases the generalization gap compared to the standard approach, it decreases it only by $0.5\%$ for most hyperparameter values, whereas, for the more global approaches, i.e., det and logdet, the generalization gap is less than $1.1\%$ in multiple cases compared to the gaps $2.87\%$ and $2.50\%$ achieved by the standard approach and the direct variant, respectively.

For the direct variant (the curves in blue), we note that the performance of the method is not sensitive to the hyperparameters, and the method achieves its best performance for low values of $\lambda$ and $\gamma$. For the det variant (the curves in orange), we note that it significantly improves the generalization ability of the model. However, there is a trade-off between the generalization gap and the final error. Emphasizing diversity and using a high weight for the diversity term significantly decreases the generalization gap, but this damages the performance of the model compared to the standard approach. For example, with $\lambda = 0.01$ and $\gamma = 10$, the generalization gap of the model is $0.9\%$ compared to $2.87\%$ of the standard. However, the test error for this model gets up to $24.42\%$ compared to $23.87\%$ for the standard. For lower values of $\lambda$, the model is able to significantly outperform the standard approach on both the test error and the generalization gap. For the logdet variant (green curves), we note that, in terms of generalization gap, the approach consistently outperforms the standard approach. Using a small value for $\lambda$, the model yields lower error rates than the standard approach. For high values of $\lambda$, the error rates become similar to the standard approach but with a lower generalization gap. This variant is not sensitive to the hyperparameter $\gamma$.

### E.3    ALL-MLP MODELS

Here, we evaluated the performance of our diversity strategy on modern attention-free, multi-layer perceptron (MLP) based models for image classification on CIFAR10. We conduct a simple experiment using two models: MLP-Mixer (Tolstikhin et al., 2021), gMLP (Liu et al., 2021) with four blocks each. For diversity strategies, i.e., ours and Decov, similar to our other experiments, the additional loss has been added on top of the last intermediate layer. The input image are resized to $72 \times 72$. We used a patch size of $8 \times 8$ and an embedding dimension of 256. all models has been trained for 100 epochs using Adam with learning rate of $0.002$, weight decay with rate $0.0001$, batch size 256. Standard data augmentation, i.e., Random horizontal Flip and random zoom with factor

Table 7: Performance of ResNet50 with different diversity strategies on ImageNet dataset with different hyper-paramters

| Method | Top-1 Test Errors | Generalization Gap |
|---|---|---|
| Standard | 23.84 | 2.87 |
| Ours direct($\gamma = 10, \lambda = 0.0001$) | 23.63 | 2.55 |
| Ours direct($\gamma = 10, \lambda = 0.001$) | 23.75 | 2.73 |
| Ours direct($\gamma = 10, \lambda = 0.005$) | 23.83 | 2.88 |
| Ours direct($\gamma = 10, \lambda = 0.01$) | 23.58 | 2.70 |
| Ours det($\gamma = 10, \lambda = 0.0001$) | 23.48 | 2.78 |
| Ours det($\gamma = 10, \lambda = 0.001$) | 23.58 | 2.77 |
| Ours det($\gamma = 10, \lambda = 0.005$) | 23.67 | 2.70 |
| Ours det($\gamma = 10, \lambda = 0.01$) | 24.47 | 0.92 |
| Ours logdet($\gamma = 10, \lambda = 0.0001$) | 23.52 | 2.58 |
| Ours logdet($\gamma = 10, \lambda = 0.001$) | 23.64 | 1.07 |
| Ours logdet($\gamma = 10, \lambda = 0.005$) | 23.88 | 2.70 |
| Ours logdet($\gamma = 10, \lambda = 0.01$) | 23.79 | 2.64 |
| Ours direct($\gamma = 1, \lambda = 0.001$) | 23.66 | 2.86 |
| Ours direct($\gamma = 50, \lambda = 0.001$) | 23.85 | 2.67 |
| Ours det($\gamma = 1, \lambda = 0.01$) | 23.77 | 2.88 |
| Ours det($\gamma = 50, \lambda = 0.01$) | 23.97 | 2.97 |
| Ours logdet($\gamma = 1, \lambda = 0.001$) | 23.65 | 2.70 |
| Ours logdet($\gamma = 50, \lambda = 0.001$) | 23.70 | 2.65 |
| Ours logdet($\gamma = 50, \lambda = 0.005$) | 23.57 | 2.78 |

$20\%$ has been used. We use $10\%$ of the training data for validation. We also reduce the learning rate by a factor of 2 if the validation loss does not improve for 5 epochs and use early stopping when the validation loss does not improve for 10 epochs. All experiments has been repeated over 10 random seeds and the average results are reported.

## E.4 TRANSFER LEARNING

For the transfer learning experiment, we use ResNet50 models pre-trained on ImageNet and we finetune them on CIFAR10 and CIFAR100. Diversity strategy has been applied in both phases. The models are trained for 20 epoch using Adam optimizer with learning rate equal 0.0001 and standard data augmentation is applied. The original images of CIFAR are preprocessed and resized to (96,96,3) in order to be adequate for ResNet50 trained on ImageNet. As it can be seen, employing a diversity strategy helps in the transfer learning context and leads consistently to lower error rates on both datasets. For example, the log variant of our approach leads to $0.94\%$ and $1.27\%$ gains on CIFAR10 and CIFAR100, respectively.

## E.5 LEARNING IN THE PRESENCE OF LABEL NOISE

To further demonstrate the usefulness of promoting diversity, we test the robustness of our approach in the presence of label noise. In such situations, standard neural network tend to overfit to the noisy sample and not generalize well to the test set. Enforcing diversity can lead to better and richer representation attenuating the effect of noise. To show this, we performed additional experiments with label noise ($20\%$ and $40\%$) on CIFAR10 and CIFAR100 using ResNet50. We use the same training protocol used for the original CIFAR10 and CIFAR100: all models are trained using stochastic gradient descent (SGD) with a momentum of 0.9, weight decay of 0.0001, and a batch size of 128 for 200 epochs. The initial learning rate is set to 0.1 and is then decreased by a factor of 5 after 60, 120, and 160 epochs, respectively. We also adopt a standard data augmentation scheme that is widely used for these two datasets (He et al., 2016; Huang et al., 2017). For all models, the additional diversity term is applied on top the last intermediate layer. For the hyperparameters: The loss weight is chosen from $\{0.00001, 0.00005, 0.0001, 0.0005, 0.001, 0.005, 0.01\}$ for both our approach and De-

cov and $\gamma$ in the radial basis function is chosen from $\{0.01, 0.1.1, 10, 50, 100\}$. For each approach, the model with the best validation performance is used in the test phase. The average errors over three random seed are reported.

### E.6 WEIGHT VS FEATURE DIVERSITY

Weight diversity is an active field of research which applies a diversity regularizer on top of the weights instead of the representation. Here, we compare the performance of two weight diversity approaches, i.e., (Yu et al., 2011; Xie et al., 2015b), against activation diversity approaches. Results are reported in 8. As it can be seen, activation-based diversity leads to superior results compared the weight-based.

Table 8: Performance of ResNet50 with different diversity strategies on ImageNet dataset

|  | Method | Top-1 Errors |
|---|---|---|
| Weight diversity | (Yu et al., 2011) | 25.08 |
|  | (Xie et al., 2015b) | 25.24 |
| Activation diversity | DeCov | 23.62 |
|  | Ours(direct) | 23.75 |
|  | Ours(det) | 23.62 |
|  | Ours(logdet) | 23.64 |

