# OpenReview forum: "Improving Neural Network Generalization via Promoting Within-Layer Diversity"
_ICLR.cc/2022/Conference — ICLR 2022 Submitted_

### Official Review · Reviewer_WnGv · 2021-11-02

**Correctness:** 3
**Technical Novelty And Significance:** 2
**Empirical Novelty And Significance:** 2
**Recommendation:** 5
**Confidence:** 3

**Main Review:**

My review on this paper is mixed. The main concern is on the theory side.

The Assumptions 1 seem way too strong so that the conclusion of Theorem 1 is not interesting any more. In particular there may be some issues with the last line of assumption in assumptions 1.


First that line of assumption is not stated clearly.

a) There is a minor notation issue on the indexing i and j. They should be m and n instead?


b) Is this an argument for all w or for a fixed w or for a randomized w? This needs to be explicitly stated.

c) From the proof I turn to believe the authors are assuming that with probability \tau the inequality holds for all ws in the last line of assumption. I would say, for a fixed w, such assumptions may make some sense. Now to prove the theorem 1 we need that assumption to hold for all ws with probability \tau. And we have an infinite number of ws. That assumption is almost as strong as the original Rademacher bound on the model generalization. As a consequence the theorem 1 is not so informative to me.


On the other hand, the empirical side does show that diversifying the within-layer output may help improve models’ performance. This is an interesting empirical observation even though I do not quite buy the authors' claim on the connection to the Rademacher bound.


**Summary Of The Paper:**

The paper proposes an interesting angle on the generalization of neural networks. Through proving a new generalization bound that has an internal distance term, the authors argue the neural networks’ generalization performance is related to the internal diversity of the neurons within each layer. As such, some regularization terms are introduced to encourage the internal ``diversity” within each layer. The effectiveness is verified empirically through experiments on a series of data sets including CIFAR and ImageNet.


**Summary Of The Review:**

Overall I feel this is a borderline paper with a mixed feedback from me. It has some interesting thoughts and empirical results with come concerns on the theoretical justifications.


#######Review after reading the authors' feedback########


I would like to thank the authors for their explanation and the revised draft. Still I feel the assumption in the theoretical justification is kind of strong even though similar assumptions have been used in some previous work. I would keep the ratings unchanged.

---

> ### Author Response · Authors · 2021-11-23
> **Response to Reviewer WnGv**
>
> **Q**:  There is a minor notation issue on the indexing i and j. They should be m and n instead? \
> **A**: Yes, thank for catching this typo. It is fixed in the updated manuscript.
>
> **Q**:  The Assumptions 1 seem way too strong so that the conclusion of Theorem 1 is not interesting any more. \
> **A**: As explained in our answer to Reviewer vc2s, in our theoretical analysis we considered a relaxed variant of the following assumption 'H*: There exists a lower bound to the distance, valid for any input x'. H* is impractical especially if the intermediate layer has ReLu activations. This is why we considered a relaxed variant of this assumption by introducing the relative probability $\tau$: 'With a high probability $\tau$,  the average pair-wise distance between the output of units, $\frac{1}{2M(M-1)} \sum_{n \neq m}^M ( \phi_n(\textbf{x}) - \phi_m(\textbf{x}))^2$ is lower bounded by $d_{min}$ for any input $\textbf{x}$'.  The probabilistic items in this assumption is the input $\textbf{x}$. From a practical point of view, the assumption made in the paper using $\tau$ allows $d_{min}$ to be strictly positive and high in practice for all Ws. Thus, it makes the theoretical findings useful. Note that this assumption is similar to the one in (Xie et al 2015b).

---

### Official Review · Reviewer_LXoK · 2021-11-03

**Correctness:** 3
**Technical Novelty And Significance:** 2
**Empirical Novelty And Significance:** 2
**Recommendation:** 3
**Confidence:** 4

**Main Review:**

Strengths:
1. This work has both a theoretical analysis of the proposed diversity regularizer and an empirical study of the effectiveness of the regularizer on existing models.

Weaknesses:
1. This work appears similar to existing diversity promoting regularizers, especially Xie et al 2015b (https://arxiv.org/pdf/1511.07110.pdf). While the diversity regularizer proposed here is applied to the neural activations instead of the weights, there seems to be no intuition/justification for why a diversity regularizer over activations is preferred over a diversity regularizer over weights, other than this sentence "... Decov, yielded superior empirical performance; however, it lacks theoretical proof. In this paper, we closed this gap ..." (page 7, first paragraph). I think this work would be stronger if it can either make a theoretical argument of why diversity regularization over activations is better, or make an empirical argument by comparing to Xie et al 2015b as well as other diversity-promoting works.

2. This work aims at regularizing neural networks, so more broadly it would be nice to compare to other regularization methods such as dropout and MC dropout.

3. Theoretically, I am not convinced that a better generalization bound always leads to a better performance, for several reasons: first, while the generalization error decreases with regularization, the approximation ability gets worse (i.e., the model cannot fit the training data as well), and this part is not analyzed in this paper. Second, the generalization bound seems to be very loose upon inspecting the constants: for example, $C_5$ in Eq. (8) is on the order of $C_1C_3$, with $C_1$ being the max norm of the input and $C_3$ being the max norm of the weight of the hidden layer. Lastly, a lower generalization error doesn't justify a regularizer. For example, if we flip the direction of the regularizer, i.e., regularize the activations to be less diverse instead of more diverse, the Rademacher complexity also decreases since the function family gets more constrained, but intuitively we don't want to use such a regularizer.

4. Empirically, this work led to "0.36%−0.54% improvement on CIFAR10 and1.86%−1.94% on CIFAR100" (page 7, under table 1), which does not seem very significant. Besides, this work proposed multiple versions of diversity regularizer and presented results for all of them, while a more reasonable thing to do is to select a version on the validation set and then present the test result of that one only. Besides, from Fig. 1, the accuracy seems to be sensitive to hyperparameters.

5. Computationally, the proposed regularizer seems expensive. For both the determinant and log determinant variants, the complexity of $O(C^3)$ with $C$ being the hidden size. I wonder if it's practical to apply this regularizer to larger networks with more hidden units.

Minor Issues:
1. "was empirically been proven to" (page 2, second paragraph)
2. Eq. (14), missing sum w.r.t. j?
3. Fig. 1, the legend appears small.

Suggestions:
1. It would be interesting to see the effectiveness of the proposed regularizer under different amounts of training data to better understand its behavior.

**Summary Of The Paper:**

This paper proposes to use a regularizer to encourage the diversity of activations within each layer of deep neural networks. The contribution of this paper is two-fold: theoretically, the authors proved that more diverse activations correspond to a lower generalization error bound due to a lower Rademacher complexity; practically, this work showed that the proposed diversity regularizer led to better performance on several image classification tasks.

**Summary Of The Review:**

Overall, this is well-structured work with both theoretical analyses and empirical results. However, this work did not provide a justification of why diversifying activations is better than diversifying weights, and it did not compare to commonly used regularization techniques such as dropout and other diversity-promoting works. Therefore, I am not recommending the acceptance of this paper.

---

> ### Author Response · Authors · 2021-11-23
> **Response to Reviewer LXoK**
>
> **Q**: I think this work would be stronger if it can either make a theoretical argument of why diversity regularization over activations is better, or make an empirical argument by comparing to (Xie et al 2015b) as well as other diversity-promoting works. \
> **A**:  Thank you for the suggestion, we have now included more motivation regarding the activation vs weight diversity. We note that weight diversity has been theoretically extensively studied before. Here, the main aim was to provide both theoretical and empirical proof that activation diversity works and improves generalization. However, we agree with the reviewer that the weight vs activation diversity is interesting. To this end, in the supplementary material we added extra experiments comparing weight-diversity to our proposed approach. The results show that activation diversity leads to better results compared to weight diversity for ResNet50.
>
> **Q**:  More broadly it would be nice to compare to other regularization methods. \
> **A**: Thank you for the suggestion, we now also add dropout results.
>
> **Q**:  Theoretically, I am not convinced that a better generalization bound always leads to a better performance. \
> **A**:  Theoretically, we derived novel generalization bounds for neural networks depending on the within-layer activation diversity, $d_{min}$. The bounds derived are inversely propositional to diversity. Thus, this motivates promoting diversity.  It is true that the bounds found, similar to most bounds in NNs [1-3], depend on several other norm-based constants, which can motivate different other approaches. For example, the bound is proportional to the weight matrix norms $C_3$ and $C_4$. This motivates weight decay and provides theoretical insights on why it works. The main scope of this work is within-layer activation diversity and hence the focus with $d_{min}$. Intuitively, our theory can be interpreted as follows: given two different models with same constant A and $C_2$ but with different $d_min$. the model with the larger $d_{min}$ has a tigher bound and thus has a better generalization. \
> $[1]$ Golowich, Noah, Alexander Rakhlin, and Ohad Shamir. "Size-independent sample complexity of neural networks." In Conference On Learning Theory, pp. 297-299. PMLR, 2018. \
> $[2]$ Neyshabur, B., Bhojanapalli, S., McAllester, D. and Srebro, N., 2017. Exploring generalization in deep learning. arXiv preprint arXiv:1706.08947. \
> $[3]$ He, F. and Tao, D., 2020. Recent advances in deep learning theory. arXiv preprint arXiv:2012.10931.
>
> **Q**:  Empirically, this work does not seem very significant. \
> **A**:  While it is true that on CIFAR10 and CIFAR100, the accuracy improvement is small, we note that it is consistent along the different models. Moreover, we note that for other tasks, namely transfer learning and label noise, learning non-redundant and diverse features is more usefull and the accuracy improvement gets larger, more than to $3\%$ in some scenarios as shown in Table 4 and 5.
>
> **Q**:   Computationally, the proposed regularizer seems expensive. For both the determinant and log determinant variants a complexity of $O(C^3)$  \
> **A**:  Note that $C$ is not the total number of all units in the model but only the number of units in the last intermediate layer, which is a relatively small in most state-of-the-art architectures, e.g.,  2048 for Wide-ResNet50 and ResNet50. To validate this, we also include the time cost of the different approaches in the updated manuscript. As it can be seen, the extra time cost is less than $0.5$%.
>
> **Q**:  Minor Issues \
> **A**:  Thanks for pointing these out. We fixed the mentioned typos in the updated manuscript.
>
> **Q**:  It would be interesting to see the effectiveness of the proposed regularizer under different amounts of training data to better understand its behavior. \
> **A**:  Thank you for the suggestion. Indeed learning diverse and non-redundant representation can be usefull in case of small amount of data. Due to number of page limit, in this paper, we focused on standard image classification, transfer learning, and learning in the presence of noise. The main aim was to motivate diversity-based approaches. However, the regularizer can be usefull beyond these tasks, e.g., learning with few examples.  We hope our results will spark further research in diversity-based approaches.

---

### Official Review · Reviewer_vc2s · 2021-11-04

**Correctness:** 3
**Technical Novelty And Significance:** 3
**Empirical Novelty And Significance:** 3
**Recommendation:** 5
**Confidence:** 4

**Main Review:**

Definition:

- In the definition of the diversity, there is a $\tau$. What does that mean? What is the probability space? Does it mean when you sample a point from the input distribution, with probability $\tau$ you have the lower bound? But you should also quantify based on the weights. Am I missing something? For instance when you have a multilayer, the events are not independent so the statement of Theorem 4 is difficult to understand for me.

- It would be nice if the authors can provide some sufficient conditions under which we can have the desired lower bound on the diversity.

Motivation behind the diversity method:
- In the definition of the diversity we want that the output of activations are not close to each other. However, is it sufficient? what if the weight vectors of the next layer basically "kill" this diversity? In general, what is the impact of the weights on the diversity?

Comparison with Dropout method:
- Empirical comparison: My understanding of the dropout is that it also has a very similar impact as your method on learning. I appreciate it if you could compare the performance of your method with the dropout method.
-Computation Comparison:
What is the difference of the dropout and your method in terms of computational cost?

Theorem 4:

What is the intuition behind the Tau^p? what is the dependence of your bound on depth and width and how does it compare with Golowich et al paper?

A Note regarding the references:

In many places in the manuscript, the citations do not match with the content.
For instance on Page 2, before definition 1, almost all of the citations are wrong. PAC learning is not from Hanneke'16. VC dimension is not from those papers you cited, and the same for Rademcher complexity.



**Summary Of The Paper:**

In this paper, the authors propose a regularization technique encourage the "activation diversity". Specifically, they design a within-layer loss that add penalty to the similar neurons with the same activation pattern. They also showed that encouraging the within-layer diversity can be used to control the Rademacher complexity of model.

**Summary Of The Review:**

Very Interesting idea but the presentation can be improved.

---

> ### Author Response · Authors · 2021-11-23
> **Response to Reviewer vc2s**
>
> **Q**: In the definition of the diversity, there is a $\tau$. \
> **A**:  We note that in our theoretical analysis we consider a relaxed variant of the following assumption 'H*: There exists a lower bound to the distance, valid for any input x'. H* is impractical especially if the intermediate layer has ReLu activations (there is a high likelihood that there exists a certain input such that all units within the layer have zero activations and, thus, $d_{min}$ would be theoretically zero or very small).  This is due to the fact that $d_{min}$ is defined over all the input space. This is why we considered a relaxed variant of this assumption by introducing the relative probability $\tau$: 'With a high probability $\tau$,  the average pair-wise distance between the output of units, $\frac{1}{2M(M-1)} \sum_{n \neq m}^M ( \phi_n(\textbf{x}) - \phi_m(\textbf{x}))^2$ is lower bounded by $d_{min}$ for any input $\textbf{x}$'.  The probabilistic items in this assumption is the input $\textbf{x}$. From a practical point of view, the assumption made in the paper using $\tau$ allows $d_{min}$ to be strictly positive and high. Thus, it makes the theoretical findings useful.
>
> **Q**:  In the definition of the diversity we want that the output of activations are not close to each other. However, is it sufficient? In general, what is the impact of the weights on the diversity? What if the weight vectors of the next layer basically "kill" this diversity?  \
> **A**:  Even thought diversity is applied to the last intermediate layer, we note that the additional loss introduced by the regularizer affects the all intermediate weights due to back-propagation. In our case we applied diversity on the last intermediate layer, so the next layer is the classifier and its weights simply define the classification boundary on top of the already learned representation. However, generally speaking it is safe to say that non-redundant features lead to better optimums and, thus, are preferred by the learning algorithm. Note that this true even for the standard approach without explicit regularizer as shown in Figure 2 of the paper, i.e., diversity is increasing along the training.
>
> **Q**: Comparison with Dropout method \
> **A**: Thank you for the interesting suggestion. In the updated manuscript, we added dropout experiments and we tested the compatibility of our regularizer with dropout on ImagetNet showing that the best performance is achieved by combining both
>
> **Q**: $\tau^p$ in Theorem 4 \
> **A**:$\tau^p$ is not '$\tau$ power $p$'. Instead, $p$ indicates the index of the layer. Note that Theorem 4 considers the case of multi-layer model where for each layer $p$, we have a relative diversity assumption and a relative constant $\tau^p$. To avoid confusion, we change the terminology to $\tau^{(p)}$.
>
> **Q**: Note regarding the references\
> **A**:  Thanks for pointing this point. We have fixed the references in the updated manuscript.

---

> > ### Comment · Reviewer_vc2s · 2021-11-29
> > **Thanks**
> >
> > I would like to thank the authors for their explanation and the revised draft. Still I think the theoretical part is quite disjoint from the empirical part. I would like to see the connection of Theorem 4 with the results of Golowich et al. I would keep the ratings unchanged.

---

### Official Review · Reviewer_dHr7 · 2021-11-05

**Correctness:** 3
**Technical Novelty And Significance:** 2
**Empirical Novelty And Significance:** 2
**Recommendation:** 3
**Confidence:** 3

**Main Review:**

Strengths:
* Theoretically-motivated and simple (in a good way) technique to encourage activation diversity.
* Empirically evaluated a wide set of benchmarks and tasks.
* The significant generalization gap reduction in Table 2 (from 2.77 -> 1.07) is intruiging and worth additional study.

Weaknesses

The mathematical section was difficult to follow for this reviewer, because the theorems and lemmas are presented with minimal description or intuition of how they were derived and connected. The paper could be improved by only describing one of the cases (e.g. single hidden-layer network) in the main text, but in greater detail, and leaving the rest to the appendix. As an example in Theorem 3, its difficult to reason about the effect of d_min on the bounds, as J appears in two terms. The explanation would benefit from a figure showing how the bounds change with d_min.

To provide a more direct link between the theory, loss term, and empirical results, the paper could be improved by visualizing the activation diversity across layers -- given that the diversity term is only applied in one layer, what is its broader effect?

Empirically, the results are mixed. On CIFAR10/CIFAR100, the apporach improves over DeCov. For ImageNet, the approach does not yield better results than previous work (DeCov) on one model, but improves on a different model. With MLP-mixer, some methods improve over DeConv, some do not. It should be noted that the DeCov results are not cited from the original paper, but the authors own implementations (because DeCov was only evaluated on AlexNet at the time).

Methodological questions:
* how is the author applying ResNet-50 on CIFAR10 and CIFAR100? The original paper (He et al, 2016) uses a slightly modified architecture in ResNet-56.
* given that DeCov results are not from the original paper, how were its hyperparameters tuned? e.g. in Table 7 in the Appendix, I see tuning of the hyperparameters for the author's method, but not for DeCov.

**Summary Of The Paper:**

The authors propose a theoretically-grounded technique to improve neural network generalization. The loss is augmented with a term that encourages units to have a more diversity in its activations. More specifically, the authors use a radial basis function to measure pairwise similarity between activations and proposed loss terms based on either the sum or the determinant of the similarity matrix.

Based on experiments in several classification models on CIFAR10, CIFAR100, and ImageNet, models trained with this additional loss term achieve higher test accuracy, and also reduce the generalization gap between train and test.

**Summary Of The Review:**

The proposed method is appealingly simple, but yields mixed results. The paper's presentation and analysis does not present a strong link between the theory and the empirical section, which leaves me to recommend not accepting the paper in its current form.

---

> ### Author Response · Authors · 2021-11-23
> **Response to Reviewer dHr7**
>
> **Q**: The mathematical section was difficult to follow for this reviewer. \
> **A**: Thank you for raising the concern. As suggested, we restructured the theoretical part to include only the single hidden layer case with more description of the proof and moved the rest to the appendix, i.e., extension to multilayer and classification case.  We also added more description clarifying how the proposed regularizer is connected to the theoretical bounds. Moreover, we added an experiment validating that our regularizer indeed improves the diversity as defined theoretically.
>
> **Q**:  Given that the diversity term is only applied in one layer, what is its broader effect? \
> **A**: Note that even thought the regularizer is applied on top of the last intermediate layer, it affects all the layers' parameters due to back-propagation.
>
> **Q**:  Empirically, the results are mixed. \
> **A**: The main contribution of the paper is that activation diversity helps. This has been shown both theoretically and empirically. All diversity-based approaches, DeCov and ours, consistently lead to improvement compared to standard approach, along multiple models, datasets, and tasks.  Compared to DeCov, we note that the det variant of our approach consistently outperforms DeCov in all experiments except on  CIFAR10 with label noise $20\%$. The logdet variant also outperforms DeCov in all experiments except the MLP-Mixer on CIFAR10 and ResNet50 on ImageNet where the gap is less $0.02\%$.
>
> **Q**:  how is the author applying ResNet-50 on CIFAR10 and CIFAR100?  \
> **A**: We use similar models as in [1,2], i.e., reducing the strides of early convolution layers. \
> $[1]$ Bansal, N., Chen, X. and Wang, Z., Can we gain more from orthogonality regularizations in training deep CNNs?. Neurips 2018 \
> $[2]$ Zhang, H., Cisse, M., Dauphin, Y.N. and Lopez-Paz, D., 2017. mixup: Beyond empirical risk minimization.,  ICLR 2018
>
> **Q**: given that DeCov results are not from the original paper, how were its hyperparameters tuned? \
> **A**:  As mentioned in Section 5.1: For the hyperparameters: The loss weight is chosen from ${0.00001,0.00005,0.0001,0.0005,0.001,0.005,0.01}$  for both our approach and Decov  based on the validation set.

---

### Official Review · Reviewer_LPWQ · 2021-11-06

**Correctness:** 3
**Technical Novelty And Significance:** 2
**Empirical Novelty And Significance:** 2
**Recommendation:** 5
**Confidence:** 4

**Main Review:**

*The purpose of the bound*

The authors derive an upper bound on Rademacher complexity. However, as a reader I am not convinced of the importance/utility of these bounds:
The bounds depend on upper bounds on norms of the weights. The norms are distribution dependent. Further, such norms (in unregularized SGD and SGD with weight decay) have been shown to grow with the amount of data (Nagarajan and Kolter ‘19). Thus it is not even clear if the bound decreases with the amount of data when considering classifiers that are obtained with the proposed algorithm. Do the norms grow with the data? If so, how fast relative to the inter-layer activation diversity term?
Demonstrating that the bound on Rademacher is tight for some distributions would be another way to demonstrate that the bounds are interesting.

The authors talk about studying generalization starting in the abstract and throughout the text, e.g.: “..we analyse the effect of the within-layer activation diversity on the generalization error bound of neural network.”.

I see multiple inaccuracies with this and related statements: the authors provide a bound, rather than study an effect of within-layer activation diversity -- the effect would be totally different on a different upper bound. And these are just upper bounds, after all.

I also wanted to note that what is actually provided in the paper is an estimation error bound, not a generalization error bound. I am guessing the authors arrived at the estimation error by starting from a Rademacher-based risk bound. But there is no mention of that. Why not state a generalization bound instead?

Here is another statement (out of several) that is very misleading and incorrect (Section 3, first sentence):

“As shown in the previous section, promoting diversity of activations within a layer can lead to tighter generalization bound and can theoretically decrease the gap between the empirical and the true risks.”

Where are the results showing that promoting diversity of activations decreases the gap between the risk and empirical risk? Also, the bounds are tighter than what? There is no comparison to any other bounds.

Just after Eq (3), f* is defined as argmin_f L(f). What class is the minimization performed over?


*How does activation diversity connect to the rest of the literature*

The authors discuss other approaches to increasing weight, activation, etc., diversity. However, since they argue that increased diversity improves generalization, it would be interesting to see a comparison to other properties that are linked to improvements in generalization. Probably one of the most recently discussed properties is the flatness of the minima. Are these complementary properties? Do they relate in any way? (see, e.g., work on Sharpness Aware minimization by Foret et al, Computing Nonvacuous Generalization Bounds by Dziugaite et al, Sharp minima work by Keskar et al., and many others)

What exactly is “Standard Training” in Section 5? Is it regularized in any way? How would the performance of the proposed algorithm compare to various other regularization methods?

*The efficiency*

In my opinion the real contribution of the paper is the algorithm, not the bound on estimation error. I do not see any discussion around the computational costs or training times of the algorithm.

*Hyperparameter settings*

The algorithm requires tuning regularization-related hyperparameters. How are those chosen exactly? Do the authors have a held-out set? Based on Fig 1, it seems that the wrong values of these hyperparameters may wipe out all the improvements in generalization presented in Table 1.

Also, the discussion of Figure 1 and more generally section 5 once again suggest that one may expect to see a smaller generalization error gap when optimizing for the inter-layer activation diversity. There is no such theoretical connection made in the paper.

How about other hyperparameters? How are those set? Are they all standard or have they been changed at all?

*Ensembling*

The authors several times relate activation diversity to functional diversity (introduction, section 4). I do not quite understand the connection. I can easily construct cases how improved activation diversity in individual classifiers would result in a poor ensemble. Can the authors explain the connection?


Minor: The loss set  in Lemma 1 is undefined.


**Summary Of The Paper:**

The authors prove an upper bound on the estimation error that depends on pairwise distances between hidden unit values per layer. The bound itself is quite likely very loose in standard deep learning settings due to the dependence on various norms of the weights. Loosely inspired by the bound, the authors then propose an optimization algorithm with a particular regularizer that encourages within-layer activation value diversity. The empirical results suggest that the proposed algorithm boosts the performance in most regimes studied (CIFAR 10, CIFAR100 trained on ResNets, Dense nets). The boost is minimal for standard datasets, but it increases when the labels are randomized. Compared to standard empirical risk minimization, a small boost in test accuracy on transfer learning tasks is also reported.

**Summary Of The Review:**

To reconsider my score, I would like to see the language around implications of the bound clarified throughout the text (in response to my comments under “the purpose of the bound”). Currently there is a huge disconnect between what has been shown and what the claims are.

Also, since the main contribution is the algorithm, I would like to see more evidence towards the algorithm potentially being adopted in practice. What are the computational costs? How do they compare? How does the algorithm compare to other recently proposed algorithms that were derived to improve generalization (like sharpness aware minimization)? How does the algorithm compare to other regularization techniques?

---

> ### Author Response · Authors · 2021-11-23
> **Response to Reviewer LPWQ**
>
> Thanks for your feedback. We appreciate your concern about several phrasings in the paper. To address that, we have updated the manuscript and incorporated many of your suggestions in the updated paper. Please, see below our detailed response and we hope it resolves any concerns you had:
>
> *The purpose of the bound:*
>
> **Q**: It is not even clear if the bound decreases with the amount of data when considering classifiers that are obtained with the proposed algorithm. \
> **A**: We note that all the bounds are inversely proportional to number of training data, i.e., $\frac{1}{N}$. This implies that the bound goes to zero as the number of training sample increases.
> Studying how our regularizer affects the norms of the learned weight distribution is interesting and can be a nice extension of this work. Here, the main aim was to motivate activation diversity as a strategy for improving generalization. We hope this will spark further research in diversity-based approaches and how they affect the properties of the learned model, e.g., weight norms.
>
> **Q**: Where are the results showing that promoting diversity of activations decreases the gap between the risk and empirical risk? Also, the bounds are tighter than what? There is no comparison to any other bounds. \
> **A** :  This is shown theoretically in Theorem 1 and empirically in Table 2. The main idea of this paper is that by promoting the diversity and penalizing redundancy between the neurons within the same layer, we can improve the generalization. The appearance of the term related to the layer output diversity, i.e., $d_{min}$, in the generalization bound in Theorems 1-4 provides theoretical support for the proposed idea. Note that we are not comparing our bound against prior bounds. We just derived a novel upperbound depending on the diversity. For example in Theorem 1, we derived a bound for the generalization gap, i.e., $  L(\hat{f}) - L(f^*) $, which is inversely proportional to $d_{min}$. This means when $d_{min}$ is higher, the bound gets smaller (tighter) and as the result the gap gets smaller. Such theoretical reasoning has been applied before, see for example [1-4]. \
> $[1]$ Xie, P., Deng, Y. and Xing, E., 2015. On the generalization error bounds of neural networks under diversity-inducing mutual angular regularization. arXiv preprint arXiv:1511.07110. \
> $[2]$ Gouk, H., Hospedales, T.M. and Pontil, M., 2020. Distance-Based Regularisation of Deep Networks for Fine-Tuning. ICLR 2021 \
> $[3]$ Cortes, C., Mohri, M. and Storcheus, D., 2019. Regularized gradient boosting. Advances in Neural Information Processing Systems. \
> $[4]$ Wan, L., Zeiler, M., Zhang, S., Le Cun, Y. and Fergus, R., 2013, May. Regularization of neural networks using dropconnect. In International conference on machine learning (pp. 1058-1066). PMLR.
>
> **Q**: How does activation diversity connect to the rest of the literature. \
> **A**: The connections to related literature has been described in the introduction and the related work section. We added the references mentioned by the reviewer. As the proposed regularizer is an activation-based diversity approach, we focused on our experimental comparisons with prior activation-based diversity approaches, i.e., 'Decov'. We also now included in the updated manuscript dropout experiments and weight-based diversity.
>
> **Q**: What exactly is “Standard Training” in Section 5? \
> **A**: For standard approach, the only difference is not using an additional diversity loss. The remaining regularizers, e.g., data augmentation, weight decay etc, are all applied as specified per-experiment. we clarified this in the updated manuscript.
>
> **Q**: I do not see any discussion around the computational costs or training times of the algorithm. \
> **A**: Computational cost is discussed in the end of Section 3.  We also now add time complexity in Section 5.1 ImageNet experiment.
>
> **Q**: The authors several times relate activation diversity to functional diversity (introduction, section 4). I do not quite understand the connection. \
> **A**: That is not related to our approach but another related line of research which considers diversity in ensemble learning.

---

> > ### Comment · Reviewer_LPWQ · 2021-11-29
> > **Significance**
> >
> > A generalization bound that has a data-dependent term in the numerator is meaningless, until you can characterize the behavior of that term as a function of data, either theoretically or empirically. For example, Nagarajan and Kolter '19 show that standard bounds with data-dependent terms increase with the amount of data despite the bound looking like 1/sqrt{N} bound.
> >
> > The authors response is not convincing me how their theory is interesting or relevant beyond loosely inspiring their regularization methods. Based on Table 3, it looks like adding the proposed regularization method on top of dropout barely improves the performance.
> >
> > Do the authors see their proposed regularization method being adopted in practice? If so, perhaps they can reiterate in what cases their regularizer brings significant performance or computational gains.

---

> > > ### Author Response · Authors · 2021-11-29
> > > **Response to Reviewer LPWQ**
> > >
> > > **Theoretical Significance:** \
> > > We agree with the reviewer that the works of (Zhang et al., 2017) or (Nagarajan and Kolter, 2019) show that uniform convergence is not optimal to theoretically understand generalization. However, this does not mean that these bounds are not useful (see for example [1,2] and discussion in Section 3.1 in [3]). This is why they are still used in the literature [4-6].  Moreover, we note that (Nagarajan and Kolter, 2019) consider only the case of over-parametrized models, while here our theory is also valid for shallow models.
> > >
> > > [1] Negrea, J., Dziugaite, G.K. and Roy, D., 2020, November. In defense of uniform convergence: Generalization via derandomization with an application to interpolating predictors. ICML 2020. \
> > > [2]  Dziugaite, G.K., Hsu, K., Gharbieh, W., and Roy, D.M., 2020. On the role of data in PAC-Bayes bounds. AISTATS 2021.  \
> > > [3] Kawaguchi, K., Kaelbling, L.P. and Bengio, Y., 2017. Generalization in deep learning. arXiv preprint arXiv:1710.05468.  \
> > > [4] Esser, P., Chennuru Vankadara, L. and Ghoshdastidar, D., Learning Theory Can (Sometimes) Explain Generalisation in Graph Neural Networks. Neurips 2021.   \
> > > [5] Dong, K., Yang, J. and Ma, T., 2021. Provable model-based nonlinear bandit and reinforcement learning: Shelve optimism, embrace virtual curvature. Neurips 2021.  \
> > > [6] HaoChen, J.Z., Wei, C., Gaidon, A. and Ma, T., 2021. Provable Guarantees for Self-Supervised Deep Learning with Spectral Contrastive Loss. Neurips 2021.
> > >
> > > **Practical Significance:** \
> > > While applying diversity leads to marginal but consistent improvement in the standard classification setting, we note that our regularizer, i.e., promoting diversity, leads to substantial improvement in other tasks, such as transfer learning (Table 5) and learning in the presence of noise (Table 6). We also think that investigating diversity might lead to better regularizers with more substantial improvement even in standard classification settings in the future.

---

### Author Response · Authors · 2021-11-23
**Rebuttal Revision Submitted**

We thank all reviewers for the time and expertise they have invested in these reviews. We have addressed all comments and improved our paper according to your suggestions. Here, we would like to point out the changes made during this revision:

  **1**- We fixed the typos and improved the overall phrasing as suggested by Reviewer LPWQ. \
  **2**- As suggested by Reviewer dHr7, we restructured the theoretical part to include only the single hidden layer case with more description of the proof and moved the rest, i.e., extension to multilayer and classification case, to the appendix. \
  **3**- We added more description clarifying how the proposed regularizer is connected to the theoretical bounds. Moreover, we added an experiment validating that our regularizer indeed improves the diversity as defined theoretically. \
  **4**- As suggested by Reviewers vc2s and LXoK, we added dropout experiments and we tested the compatibility of our regularizer with dropout on ImagetNet. \
  **5**- In the appendix, we added an extra experiment comparing weight diversity to activation diversity.

---

### Decision · Program_Chairs · 2022-01-20

**Decision:**

Reject

**Comment:**

This work proposes an approach to encourage within-layer diversity in neuron activations, and derive a generalization bound meant to motivate their approach.

None of the reviewers support the acceptance of this work, despite the authors' detailed rebuttals, with the majority of reviewers confirming their preference for rejection following the author response. Many raised concerns regarding the value of the accompanying theory. The empirical results demonstrated by the proposed regularizer were also not judged to be sufficiently compelling to compensate for the shortfall on the theory side.

I unfortunately could not find a good reason to dissent from the reviewers majority opinion, and therefore also recommend rejection at this time.